# KBVQ-MoE: KLT-guided SVD with Bias-Corrected Vector Quantization for MoE Large Language Models

**Zukang Xu**[1*]   **Zhixiong Zhao**[1,2*†]   **Xing Hu**[1]   **Zhixuan Chen**[1]   **Dawei Yang**[1‡]

[1] Houmo AI        [2] Nanyang Technological University

## ABSTRACT

Mixture of Experts (MoE) models have achieved great success by significantly improving performance while maintaining computational efficiency through sparse expert activation. However, their enormous parameter sizes and memory demands pose significant challenges for deployment in resource-constrained environments. Vector Quantization (VQ) offers a promising approach for ultra-low-bit compression in Large Language Models (LLMs) by constructing and leveraging a codebook—where weight vectors are mapped to the most similar discrete codewords within the codebook. However, its direct application to MoEs suffers from significant performance degradation caused by two critical obstacles: (1) redundant representation among experts leads to VQ repeatedly quantizing similar representations for each expert, resulting in inefficient utilization of the limited codebook capacity; and (2) cumulative outputs bias is amplified by experts aggregation in MoE layers, leading to distributional shifts in the quantized outputs. To this end, we propose KBVQ-MoE, a novel VQ framework to enhance extremely low-bit quantization for MoE-based LLMs. KBVQ-MoE integrates two novel techniques: (1) Input-driven redundancy elimination, where a Karhunen–Loève Transform (KLT) guided singular value decomposition (SVD) extracts and shares dominant weight components across experts. (2) Bias-corrected output stabilization, where vector quantization is applied to expert-specific (i.e., non-redundant) representations and the quantized outputs are corrected with channel-wise affine compensation. Experiments on various MoE LLMs demonstrate that our KBVQ-MoE preserves accuracy substantially better than existing quantization methods. For instance, 3-bit quantization of Qwen1.5-MoE-A2.7B achieves an average accuracy of 67.99, nearly identical to the FP16 baseline of 68.07, underscoring the potential of KBVQ-MoE for efficient deployment on edge devices and other resource-constrained platforms.

## 1 INTRODUCTION

Mixture-of-Experts (MoE) models have recently achieved state-of-the-art performance in natural language processing (NLP) (OpenAI, 2023; Kimi Team, 2025; DeepSeek-AI et al., 2024; Jiang et al., 2024; Yang et al., 2025). By activating only a small subset of experts through a gating mechanism, MoE enables near-linear scaling of capacity with the number of experts while keeping inference cost manageable. However, the growth in expert count substantially increases parameter storage and memory bandwidth requirements. For instance, Qwen3-Next-80B-A3B(Alibaba Cloud, 2025) requires more than 160GB of GPU memory under FP16 inference. These extreme resource requirements make deployment on edge devices largely infeasible. Post-Training Quantization (PTQ) has emerged as a promising solution for compressing LLMs without the need for retraining (Hu et al., 2024; Frantar et al., 2022). A typical subcategory of PTQ is Scalar Quantization (SQ), which represents each weight independently by mapping it to a discrete value from a lower bit-width set.

---

[*]Equal contribution

[†]This work was conducted during his internship at Houmo AI

[‡]Corresponding author

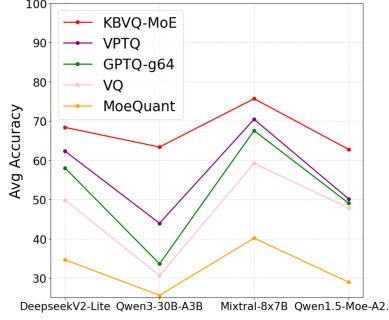

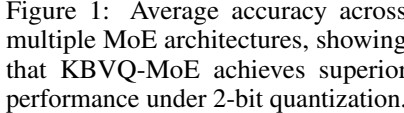

Figure 1: Average accuracy across multiple MoE architectures, showing that KBVQ-MoE achieves superior performance under 2-bit quantization.

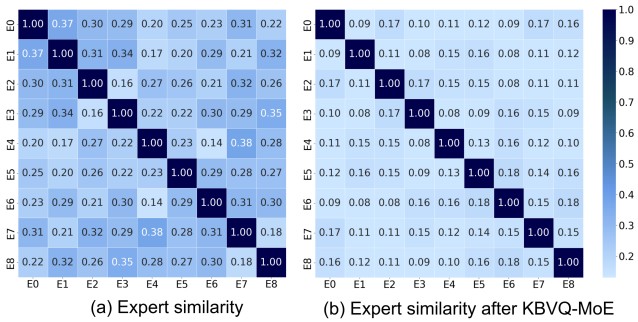

Figure 2: Similarity of expert outputs before and after redundancy elimination by KBVQ-MoE.

SQ performs well from medium to high bit-width ($\geq 4$ bits) (Xu et al., 2025; Sun et al., 2024; Zhao et al., 2025), but its representational capability drops sharply at extremely low bit-width ($\leq 3$ bits), leading to significant accuracy degradation. In contrast, Vector Quantization (VQ), another subcategory of PTQ, shows strong potential for ultra-low-bit dense LLM quantization(Yue et al., 2025; Liu et al., 2024; Tseng et al., 2024b). This advantage is realized mainly through leveraging a predefined codebook—where weight vectors are mapped to the most similar discrete codewords within the codebook—thus significantly reducing the data volume while maintaining an acceptable level of information retention.

However, directly applying VQ to MoE architectures suffers from significant performance degradation caused by two key obstacles. ❶ **Redundant representation among experts.** MoE experts often capture similar feature patterns(Du et al., 2025; Sankar & Dimitri, 2025; Omi et al., 2025; Gu et al., 2025; Li et al., 2025a), resulting in substantial parameter redundancy. As shown in Fig. 2(a), experts within the same layer produce highly similar outputs for identical activations, reflecting their overlapping functional roles. This redundancy wastes quantization capacity and prevents limited codebook resources from being concentrated on expert-specific (i.e., non-redundant) representations. ❷ **Cumulative and amplified outputs bias.** Quantization errors accumulate across layers, resulting in biased layer outputs. In MoE architectures, this bias becomes more pronounced because expert aggregation further amplifies it, leading to more severe distributional shifts than in dense LLMs. As shown in Fig.3, both the mean and variance of outputs shift after quantization. When biased

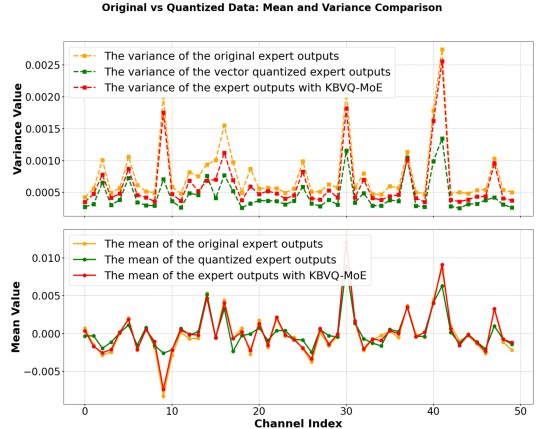

Figure 3: Distributional Shifts in Qwen3-30B-A3B Layer 20 Outputs: (Top) Per-channel Mean Comparisons (FP, Direct VQ, KBVQ-MoE); (Bottom) Per-channel Variance Comparisons (FP, Direct VQ, KBVQ-MoE).

outputs from multiple experts are aggregated through gating, the bias is amplified and propagates across layers, leading to distributional drift and degraded model performance.

To this end, we propose KBVQ-MoE, the first VQ framework tailored to MoE architectures. KBVQ-MoE is built on two efficient innovations: ❶ **Input-driven redundancy elimination(IDRE)**. First, we employ the Karhunen–Loève Transform (KLT) to align expert weights with input activation statistics, thereby mapping them into a common latent space, referred to here as a unified representation (see Eq. 3). Next, we apply SVD to this unified representation in order to extract the dominant shared representation, which are retained at full precision, making the remaining expert-specific representations easier to quantize effectively. As illustrated in Fig. 2(b), after redundancy elimination

the outputs of experts exhibit much lower similarity compared to Fig. 2(a), validating the effectiveness of redundancy elimination. ❷ **Bias-corrected output stabilization(BCOS)**. We apply vector quantization only to expert-specific(i.e., non-redundant) representations and stabilize the outputs of quantized experts through lightweight linear scaling and bias correction. As a result, with proposed IDRE and BCOS, KBVQ-MoE provide an effective solution for ultra-low-bit quantization in MoE LLMs, achieving both efficient codebook utilization and stable output distributions.

We conducted experimental evaluations of the proposed method on a variety of MoE LLMs, including Qwen1.5-MoE-A2.7B(Qwen Team, 2024), Qwen3-30B-A3B(Yang et al., 2025), Mixtral-8x7B(Jiang et al., 2024) and DeepseekV2-Lite(DeepSeek-AI et al., 2024). As shown in Fig. 1, KBVQ-MoE consistently outperforms existing scalar and vector quantization methods under the same compression ratio, with particularly strong gains in ultra-low precision settings. For example, at 2-bit quantization on the Qwen3-A3B-30B, our method improves perplexity by 6 and raises average accuracy by nearly 10%, demonstrating its potential for deploying MoE LLMs on resource-constrained devices such as edge platforms.

The main contributions of this paper are as follows:

- We identify two key challenges that arise when vector quantization is applied to MoE-based LLMs: the waste of codebook resources caused by common redundant representation among experts, and outputs bias in quantization errors exacerbated by expert aggregation.
- We propose the KBVQ-MoE framework, which integrates Input-driven redundancy elimination and Bias-corrected outputstabilization.
- Both theoretical analysis and experimental results demonstrate the effectiveness of the KBVQ-MoE method. It exhibits significant advantages over existing methods on models such as the Qwen series and Mixtral, and even achieves near-floating-point accuracy performance under 2-bit quantization.

## 2 RELATED WORK

### 2.1 POST-TRAINING QUANTIZATION (PTQ)

**Scalar Quantization (SQ)** assigns an independent scaling factor and zero point to each tensor (e.g., layer weight matrix), mapping continuous values to discrete integers. Methods such as GPTQ (Frantar et al., 2022) and GPTAQ (Li et al., 2025b) leverage Hessian information to optimize the error propagation path, while I-LLM (Hu et al., 2024) achieves quantization of large models through smoothing and full integer inference approximation. Approaches like Quarot (Ashkboos et al., 2024) and OSTQuant (Hu et al., 2025b) incorporate orthogonal transforms to improve quantization efficiency. However, SQ suffers from representational bottlenecks in ultra-low bit-width scenarios ($\leq 4$ bits), making it difficult to balance the compression ratio and accuracy of MoE LLMs.

**Vector Quantization (VQ)** clusters weight vectors into shared codebooks and leverages structural redundancy to achieve higher compression ratios (Gersho, 1979). GPTVQ (Van Baalen et al., 2024) combines expectation-maximization with error feedback to optimize codebooks; VPTQ (Liu et al., 2024) and AQLM (Egiazarian et al., 2024) employ residual quantization to reduce error accumulation; PCDVQ (Yue et al., 2025) achieves efficient quantization by decoupling the magnitude and direction of vectors; QuIP# (Tseng et al., 2024a) and QTIP (Tseng et al., 2024b) utilize geometric transformations to improve error distribution. These studies have demonstrated the superiority of VQ at extremely low bit-widths for dense LLMs. Nevertheless, when existing VQ techniques are directly applied to MoE LLMs, they fail to account for the unique structural information of MoE, resulting in suboptimal compression performance.

### 2.2 MOE LLMS COMPRESSION METHODS

Research on model compression for MoE architectures is still in its infancy. Most methods directly adopt general model compression techniques, lacking special consideration for the structural characteristics of MoE gating and multi-expert architectures. EAC-MoE (Chen et al., 2025) reduces parameters by pruning redundant experts, but pruning causes irreversible loss of functionality. D2-MoE (Gu et al., 2025) and SubMoe (Li et al., 2025a) decompose expert weights into low-rank factors,

yet their compression ratios are limited by rank constraints, failing to meet the demands of extreme resource-constrained scenarios. MoEQuant (Hu et al., 2025a) uses routing statistics to balance the contributions of each expert during calibration, but its performance remains unsatisfactory under quantization of $\leq 4$ bits.

In summary, existing MoE LLMs compression methods lack a collaborative optimization mechanism that integrates the statistical characteristics of input activations—they neither fully exploit input-related common patterns shared across experts nor specifically correct distribution shifts caused by quantization errors of experts. This makes it difficult for them to balance model accuracy at high compression ratios. The KBVQ-MoE framework proposed in this paper is designed to fill this gap, significantly improving the quantization performance of MoE LLMs through input-driven redundancy elimination and bias-corrected output stabilization.

## 3 PRELIMINARIES

**Mixture-of-Experts.** Mixture-of-Experts (MoE) architectures modify standard Transformer layers by replacing conventional feed-forward (MLP) modules with specialized MoE modules, enabling efficient scaling of model capacity while maintaining computational efficiency(Jacobs et al., 1991; Jordan & Jacobs, 1994). MoE layer typically comprises two types of components: a set of $m$ shared experts ($\{E_1^s, \ldots, E_m^s\}$), a set of $n$ routing experts ($\{E_1^r, \ldots, E_n^r\}$), and a gating network (or router network) that determines which experts process each input token.

Each expert—whether shared or routing—functions as a lightweight feed-forward sub-network (effectively a compact MLP), with shared experts designed to handle general patterns across inputs and routing experts specialized for specific input subsets. When processing an input hidden state $\boldsymbol{x} \in \mathbb{R}^d$, the MoE layer operates through a structured sequence:

1. Gating Score Calculation: The gating network computes weights (or affinities) $g_i(\boldsymbol{x})$ for all experts, quantifying how well each expert aligns with the input $\boldsymbol{x}$. These weights reflect the probability of routing $\boldsymbol{x}$ to the corresponding expert.

2. Expert Selection: For routing experts, only the top $k$ experts with the highest affinities are selected, denoted by the subset $\mathcal{K} = \text{top}k\left(\{g_j(\boldsymbol{x}) \mid j \in \{1, \ldots, n\}\}\right)$. In contrast, all shared experts are typically utilized to preserve general representational capacity.

3. Weighted Output Computation: The final output $\boldsymbol{y}$ of the MoE layer is a weighted sum of outputs from the selected experts, where the weights are the corresponding gating scores. Mathematically, this is formalized as:

$$\boldsymbol{y} = \underbrace{\sum_{i=1}^{m} E_i^s(\boldsymbol{x})g_i(\boldsymbol{x})}_{\text{shared experts}} + \underbrace{\sum_{j\in\mathcal{K}} E_j^r(\boldsymbol{x})g_j(\boldsymbol{x})}_{\text{top } k \text{ routing experts}}, \tag{1}$$

While this formulation captures the core mechanism, specific designs vary across architectures—for example, in how gating scores are computed, how experts are structured (e.g., depth, activation functions), or how the balance between shared and routing experts is tuned.

## 4 METHOD

This section introduces the proposed KLT-guided SVD with Bias-Corrected Vector Quantization for MoE Large Language Models (KBVQ-MoE). The framework is built on two innovations that align with the challenges identified in Section 1: (1) **Input-driven Redundancy Elimination (IDRE)**, where the Karhunen–Loève Transform (KLT) maps expert weights into a unified representation aligned with input statistics, and singular value decomposition (SVD) is then applied to extract dominant shared components retained at full precision, leaving expert-specific (i.e., non-redundant) representations that are more amenable to quantization; (2) **Bias-Corrected Output Stabilization (BCOS)**, where vector quantization is applied only to the expert-specific representations, and lightweight linear scaling with bias correction is used to stabilize outputs and mitigate distributional

shifts. The overall workflow of KBVQ-MoE is mathematically formulated in Eq.2, while a detailed algorithmic flow is provided in Appendix 1.

$$W \xrightarrow[\text{KLT+SVD}]{\text{IDRE}} \underbrace{W_{\text{share}}}_{\text{Shared Part}} + \underbrace{W_{\text{quant}}}_{\text{Specific Part}} \xrightarrow[\text{Bias Correction}]{\text{BCOS}} \underbrace{W_{\text{share}}}_{\text{no quant}} + \underbrace{W_{\text{quant}}}_{\text{vector quant}} + \underbrace{(s, b)}_{\text{scale\&bias.}} \qquad (2)$$

## 4.1 INPUT-DRIVEN REDUNDANCY ELIMINATION(IDRE)

In MoE architectures, experts often exhibit substantial redundancy, as their weights encode overlapping mappings for identical activations. This redundancy wastes quantization capacity and prevents the limited codebook from focusing on expert-specific representations. To address this issue, we introduce **Input-driven Redundancy Elimination (IDRE)**, which leverages the statistical characteristics of model inputs to construct a unified representation of expert weights. By isolating shared components and retaining them at full precision, IDRE reduces redundancy and improves the efficiency of codebook utilization for subsequent quantization. IDRE is applied uniformly to the MLP weights of all experts in each MoE layer—including both shared and routing experts. The detailed transformation of the MoE structure before and after applying IDRE is illustrated in Fig. 4. This procedure consists of the following steps:

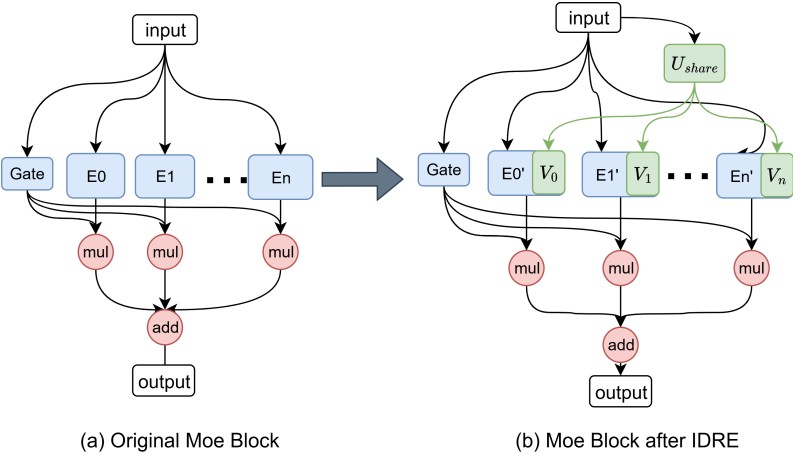

(a) Original Moe Block          (b) Moe Block after IDRE

Figure 4: The comparison of structural changes in the MoE (Mixture-of-Experts) structure before and after IDRE.

**Step 1: KLT Decomposition of Input Activations: Constructing the Input Coherence Space.** To reduce redundancy, we begin by applying the Karhunen–Loève Transform (KLT) to the input activations in order to construct a coherence basis that captures their dominant directions. Let the input activation matrix of the expert layer be $X \in \mathbb{R}^{b \times ic}$, where $b$ denotes the number of samples and $ic$ is the input dimension. The procedure is as follows:

1. The covariance matrix is computed as $C_X = \frac{1}{B-1} X^T X \in \mathbb{R}^{ic \times ic}$.

2. An eigenvalue decomposition is performed on $C_X$: $C_X = \left(U_{\text{KLT}} \Lambda_{\text{KLT}}^{\frac{1}{2}}\right)^T \left(U_{\text{KLT}} \Lambda_{\text{KLT}}^{\frac{1}{2}}\right)$. where $U_{\text{KLT}} = [u_1, \ldots, u_{ic}]$ is an orthonormal basis, and $\Lambda_{\text{KLT}} = \text{diag}(\lambda_1 \geq \cdots \geq \lambda_{ic})$ is a diagonal matrix of eigenvalues. Here, $u_j$ represents the j-th coherence direction of the input, and $\lambda_j$ denotes the energy magnitude of this direction.

3. Based on this, we construct the input coherence basis: $U_X = U_{\text{KLT}} \Lambda_{\text{KLT}}^{1/2}$ The column vectors of the resulting $U_X$ are sorted by input energy magnitude, forming an orthogonal coordinate system based on input statistics. This facilitates prioritizing the extraction of common weight structures related to high-energy input directions in subsequent steps.

A detailed theoretical derivation linking this decomposition to the minimization of quantization error is provided in Appendix A.2.

**Step 2: Mapping Weights to the Input Coherence Space.** To ensure that weight analysis is guided by the dominant directions of the input, we project the original weight matrix $W \in \mathbb{R}^{oc \times ic}$ onto the input coherence basis obtained in Step 1: $\hat{W} = WU_X \in \mathbb{R}^{oc \times ic}$, where $oc$ denotes the output channel dimension and $ic$ denotes the input channel dimension. In this transformed representation, each column of $\hat{W}$ corresponds to the mapping of a specific coherence direction of the input to the output space. The $j$-th column, in particular, represents how the weights project along the $j$-th most energetic input direction $u_j$. This alignment establishes a direct connection between weight structures and input characteristics, which facilitates the subsequent extraction of shared and expert-specific components.

**Step 3: Extracting Input-Coherent Common Structures: Precise Redundancy Elimination.** The common structures of MoE experts manifest as cross-expert shared weight patterns in the input coherence space (e.g., similar mappings for the high-energy direction $u_1$). To capture these patterns, we concateniate the mapped weights of all $n$ experts($\hat{W}^{(i)}$ denotes the i-th expert)—along their output channel dimension—into a $(n \cdot oc) \times ic$-dimensional unified representation:

$$\bar{W} = \begin{bmatrix} \hat{W}^{(1)} \\ \hat{W}^{(2)} \\ \vdots \\ \hat{W}^{(n)} \end{bmatrix} = \begin{bmatrix} W^{(1)}U_X \\ W^{(2)}U_X \\ \vdots \\ W^{(n)}U_X \end{bmatrix} \in \mathbb{R}^{(n \cdot oc) \times ic} \tag{3}$$

We then apply SVD to $\bar{W}$ in order to extract the dominant shared representation:$\bar{W} = \left( U\Sigma V^T \right)^T$.

1. $U \in \mathbb{R}^{ic \times k}$ denotes the shared left singular vectors (shared mapping directions across experts, where $k$ is the number of retained dominant singular values, focusing on input-coherent common patterns); $\Sigma = \mathrm{diag}(\sigma_1, \ldots, \sigma_k)$ is the singular value matrix (the larger $\sigma_j$ is, the more prominent the corresponding dominant shared representation).

2. The common structures in the input coherence space are mapped back to the original weight space via the inverse transformation of the input coherence basis ($U_X^{-1}$):

$$U_{\mathrm{share}} = U^T \cdot U_X^{-1} \in \mathbb{R}^{k \times ic}. \tag{4}$$

$V \in \mathbb{R}^{(n \cdot oc) \times k}$ represents the right singular vectors, which are partitioned by expert into:

$$V = \left[ \Sigma_k V^{(1)^T}, \Sigma_k V^{(2)^T}, \ldots, \Sigma_k V^{(n)^T} \right]^T, \tag{5}$$

where $V^{(i)} \in \mathbb{R}^{oc \times k}$ is the private right singular vector for expert $i$.

3. The truncated low-rank components $U_{\mathrm{share}}$ and $V$ obtained from SVD are retained in full precision to preserve the fidelity of shared and expert-specific representations.

Thus, IDRE explicitly decouples shared structures (kept at full precision) from expert-specific representations (subject to quantization), enabling more efficient use of codebook capacity. A rigorous spectral characterization of this decomposition, including its optimality and error bounds, is provided in Appendix A.3.

## 4.2 BIAS-CORRECTED OUTPUT STABILIZATION(BCOS)

While IDRE reduces redundancy and improves codebook utilization, quantization of the remaining expert-specific weights $W_{quant}$ still introduces cumulative bias that distorts output distributions. To mitigate this problem, BCOS stabilizes the outputs through vector quantization followed by lightweight channel-wise affine compensation.

**Step 1: Vector Quantization of Expert-Specific Weights.** We perform VQ quantization on the remaining expert-specific weights $W_{\mathrm{quant}}$ in the original space: partition $W_{\mathrm{quant}}$ into subvectors $z \in \mathbb{R}^d$, map these subvectors to indices$q = \arg\min_j \|z - c_j\|^2$ via a codebook $C = \{c_1, ..., c_K\}$, and the quantized result is $z_q = c_q$. The quantized matrix is denoted as: $W_{\mathrm{quant,VQ}} \in \mathbb{R}^{oc \times ic}$. The compressed weight is the sum of the shared component and the quantized component:

$$W_{\mathrm{VQ}} = W_{\mathrm{share}} + W_{\mathrm{quant,VQ}} \tag{6}$$

The quantization error, arising solely from the quantization of the specific component, is defined as $\epsilon = W_{\text{VQ}} - W$.

**Step 2: Channel-Wise Bias Correction.** As the number of layers increases, the quantization error $\epsilon$ leads to distributional shifts in the outputs. we apply channel-wise affine compensation to align the statistics of quantized outputs with those of the original model. The corrected output of the linear layer is formulated as

$$\mathbf{y}_{\text{corr}} = W_{\text{VQ}}x + s \odot (W_{\text{VQ}}x) + b = (s+1) \odot (W_{\text{VQ}}x) + b, \tag{7}$$

where $s \in \mathbb{R}^{oc}$ is a channel-wise scaling factor; $b \in \mathbb{R}^{oc}$ is a channel-wise bias term; $\odot$ denotes element-wise multiplication. Here, $W_{\text{VQ}}x$ is the original quantized output. We choose $s$ and $b$ to ensure that the corrected output $\mathbf{y}_{\text{corr}}$ matches the mean and variance of the full-precision output $\mathbf{y} = W\mathbf{x}$. Following unbiased estimation in the sense of Minimum Mean Square Error (MMSE) (see Appendix A.4), the optimal parameters are approximated as

$$s_j \approx \frac{\sigma_{y_j}}{\sigma_{\hat{y}_j}} - 1, \qquad b_j = \mu_{y_j} - (1 + s_j)\mu_{\hat{y}_j}. \tag{8}$$

where $\sigma$ and $\mu$ denote the standard deviation and mean, respectively, and both operations are channel-wise. In other words, each channel is scaled by the ratio of standard deviations between the original and quantized outputs, and shifted by the corresponding mean difference. After this correction, both the mean and variance of each channel are aligned with the full-precision baseline, effectively eliminating the distributional shift caused by quantization. This correction introduces only $2oc$ additional parameters per layer and requires only lightweight per-channel scaling and shifting operations during inference, resulting in negligible computational and storage overhead.

## 5 EXPERIMENT

**Models and Datasets.** To comprehensively verify the performance effectiveness and scenario applicability of the KBVQ-MoE framework, we conducted experiments on multiple sets of representative Mixture-of-Experts (MoE) models and diverse evaluation tasks. For the selection of evaluation models, we prioritized currently mainstream pre-trained MoE LLMs to ensure the reference value of the experimental results. Specifically, these models include the Qwen series (e.g., Qwen1.5-MoE-A2.7B (Qwen Team, 2024), Qwen3-30B-A3B (Yang et al., 2025)), DeepseekV2-Lite (DeepSeek-AI et al., 2024), and Mixtral (Jiang et al., 2024). In terms of the design of evaluation tasks and datasets, we focused on natural language reasoning and understanding benchmarks, and comprehensively examined the framework's performance through multi-dimensional metrics: - On the one hand, we tested the perplexity (ppl) of the language model (LLM) output with a sequence length of 4096 on the WikiText2 dataset (Merity et al., 2016), to measure the coherence of language generation and semantic modeling capability of the compressed model; - On the other hand, we evaluated the model accuracy on 7 zero-shot datasets, including Arc-Challenge (Clark et al., 2018), Arc-Easy (Boratko et al., 2018), HellaSwag (Zellers et al., 2019), LAMBADA-openai (Paperno et al., 2016), LAMBADA-standard (Paperno et al., 2016), PIQA (Bisk et al., 2020), and WinoGrande (Sakaguchi et al., 2021). This was done to comprehensively verify the generalization capability of the compressed model across different task scenarios.

**Baselines.** To demonstrate the effectiveness and superiority of our method, we compare KBVQ-MoE with a range of well-validated excellent methods, such as Round To Nearest (RTN) and GPTQ (Frantar et al., 2022), the MoEQuant (Hu et al., 2025a) (proposed specifically for MoE quantization), as well as VQ. We respectively compare the accuracy differences of these methods under ultra-low bit-widths ($2\tilde{3}$ bits), with their compression ratios evaluated based on actual storage usage.

**Implementation Details.** All experiments were conducted on NVIDIA RTX A6000 GPU. The calibration dataset used in our experiments was sampled from the Red_Pajama dataset (Together Computer, 2023): we fixed the random seed to 42 and randomly sampled 256 data samples with a sequence length of 4096 from the Red_Pajama dataset, which served as the calibration set for the IDRE and BCOS methods in this paper. In the IDRE process, we set the truncated rank$k$ of KLT-SVD to 1/128 of the full rank, and under this configuration, the average number of parameters increased

Table 1: Comparison of average accuracy between the KBVQ-MoE method and other quantization methods on various different moe models

| Bit | Method | Qwen1.5-MoE-A2.7B | | Qwen3-30B-A3B | | Mixtral-8x7B | | DeepseekV2-Lite | |
| --- | --- | --- | --- | --- | --- | --- | --- | --- | --- |
| | | W2 ($\downarrow$) | Avg Acc ($\uparrow$) | W2 ($\downarrow$) | Avg Acc ($\uparrow$) | W2 ($\downarrow$) | Avg Acc ($\uparrow$) | W2 ($\downarrow$) | Avg Acc ($\uparrow$) |
| 16 | FP16 | 7.22 | 68.07 | 8.70 | 70.24 | 3.88 | 78.57 | 5.92 | 70.68 |
| 2 | RTN | 638509 | 25.64 | 765922 | 25.89 | 274952 | 25.27 | 174653 | 25.12 |
| | GPTQ | 12.69 | 49.07 | 438.42 | 33.06 | 5.69 | 67.56 | 8.49 | 58.04 |
| | MoeQuant | 583542 | 34.64 | 37465 | 28.94 | 13.43 | 40.16 | 25893 | 25.59 |
| | VQ | 26.95 | 47.84 | 115.30 | 30.61 | 5.99 | 59.22 | 10.96 | 49.85 |
| | KBVQ-MoE | **9.61** | **62.78** | **11.87** | **63.37** | **5.39** | **75.69** | **7.94** | **63.10** |
| 3 | RTN | 116689 | 25.68 | 68657 | 25.84 | 45136 | 25.80 | 97.75 | 33.32 |
| | GPTQ | **7.58** | 66.36 | 11.32 | 63.51 | 4.17 | 77.43 | 6.98 | **68.89** |
| | MoeQuant | 7.87 | 66.79 | 24.85 | 37.32 | 5.45 | 72.21 | 7.52 | 66.34 |
| | VQ | 11.47 | 55.94 | 18.72 | 52.06 | 5.52 | 73.98 | 7.94 | 62.26 |
| | KBVQ-MoE | 7.74 | **67.99** | **9.26** | **69.09** | **4.07** | **78.35** | **6.95** | 68.73 |

by SVD redundancy extraction is only 0.12. In this paper, the length of the vector quantization vector length is set to 4. This selection will reduce the occupation of the codebook to a low level. Meanwhile, the k-means algorithm adopted in the vector quantization process uses kmeans++ for initialization and sets the iteration number to 100, which helps balance the stability of codebook generation and the efficiency of quantization computation. All evaluations were performed using the open-source LM-Evaluation-Harness toolkit (Gao et al., 2021).

**Main results.** Table 1 presents the results of direct comparisons between our method and other approaches. Overall, our method demonstrates significant advantages across different models and quantization bit-widths, achieving average accuracy (Avg Acc) close to full precision (FP16) while maintaining low error. Particularly in low-bit quantization scenarios ($\leq$ 3 bits), our method exhibits notable robustness and generalization capability. For Qwen models, take Qwen3-30B-A3B(Yang et al., 2025) under 2-bit quantization: its PPL decreases by nearly 6 points, and Avg Acc increases by more than 10 points. For Qwen1.5-MoE-A2.7B(Qwen Team, 2024) and Mixtral-8x7B(Jiang et al., 2024) under 3-bit quantization, their Avg Acc reaches 67.99 and 78.35, respectively, which are nearly identical to the FP16 precision. Detailed accuracy results for each dataset and model are provided in Appendix A.7

Table 2: Ablation studies on the pre-process and post-process of KBVQ-MoE respectively

| Model | Bit | IDRE | BCOS | W2 | ARC-E | ARC-C | HE | PIQA | WI |
| --- | --- | --- | --- | --- | --- | --- | --- | --- | --- |
| Qwen3-30B-A3B | FP16 | - | - | 8.70 | 79.25 | 56.40 | 59.60 | 80.30 | 70.48 |
| | 3 | ✗ | ✗ | 18.72 | 57.83 | 40.87 | 63.23 | 71.82 | 57.70 |
| | | ✓ | ✗ | 11.67 | 71.35 | 50.55 | 73.51 | 77.75 | 66.92 |
| | | ✗ | ✓ | 14.32 | 65.49 | 47.33 | 68.37 | 75.52 | 60.42 |
| | | ✓ | ✓ | **9.26** | **77.27** | **53.24** | **75.53** | **78.89** | **70.01** |

**Ablation Experiments.** As shown on Table 2. In the 3-bit quantization of Qwen3-30B-A3B, our complete scheme (IDRE + BCOS) achieves the best performance across the board: the perplexity (ppl) on Wikitext2 decreases to 9.26, representing a 50.5% reduction compared to the unprocessed baseline (18.72) — this is a 20.6% decrease compared to the IDRE-only scheme and a 35.3% decrease compared to the BCOS-only scheme. Additionally, comprehensive improvements are observed across five benchmark tasks. From the ablation of individual modules, it is evident that IDRE is the primary source of performance gain, while BCOS further corrects residuals and unleashes a synergistic effect on this foundation. Their combination enables a significant leap in overall accuracy, demonstrating that the proposed two innovations can approach full-precision performance under extremely low bit-widths. This validates its strong robustness and practical value for real-world deployment.

Table 3: Ablation on using KLT to guide SVD for expert redundancy

| Model | Bit | Method | W2 | ARC-E | ARC-C | HE | PIQA | WI |
| --- | --- | --- | --- | --- | --- | --- | --- | --- |
| Qwen1.5-MoE-A2.7B | 2 | SVD+VQ | 14.03 | 68.21 | 43.59 | 67.32 | 76.60 | 66.80 |
| | | KLT-SVD+VQ | 11.87 | 70.33 | 47.61 | 67.49 | 76.66 | 66.46 |

In IDRE, we use the Karhunen–Loève Transform (KLT) to extract the input coherence basis for redundancy removal across experts, which enables more direct redundancy extraction. In Table 3, we compare the direct performance between redundancy extraction without input coherence and that with KLT-based input coherence. Clearly, the IDRE scheme yields greater accuracy improvement: specifically, on the Qwen1.5-MoE-A2.7B model, the perplexity (ppl) on Wikitext2 decreases by more than 2 points.

For the IDRE, we conducted ablation studies to determine the optimal ratio of the truncated rank $k$ of SVD to the full rank $n$. Table 4 presents the ablation results of Qwen3-30B-A3B under 2-bit quantization: without IDRE, the model performs poorly, with a perplexity (ppl) of 15.3 on Wikitext2. However, after applying SVD truncation to pre-extract expert redundancy, the ppl drops sharply to 11.87. Further increasing

Table 4: Ablation study on the rank of SVD truncation in the Pre-Process procedure

| Model | Bit | k/n | W2 |
|---|---|---|---|
| | 2.01 | 0.0 | 15.30 |
| Qwen3-30B-A3B | 2.08 | 1/128 | 11.87 |
| | 2.11 | 1/64 | 11.30 |
| | 2.20 | 1/32 | 11.01 |

the truncated rank, performance improvements become marginal, while the average bit-width increases instead. Therefore, we recommend setting the truncated rank to $1/128$ of the full rank.

To further validate the effectiveness of our approach on MoE architectures, we integrate our IDRE and BCOS modules with different vector quantization baselines, treating them as a general plug-in component for MoE quantization. This design demonstrates that our method is both versatile and efficient, as it can be seamlessly combined with existing quantization techniques to enhance their performance. As shown in Table 5, on Qwen1.5-MoE-A2.7B(Qwen Team, 2024) applying our method to GPTVQ(Van Baalen et al., 2024) under 3-bit quantization yields nearly a 30% ppl improvement, while under 2-bit quantization with VPTQ(Liu et al., 2024), our method still achieves more than a 15% performance gain.

Table 5: Experimental Results of Different VQ Methods Integrated with KBVQ-MoE for Qwen1.5-Moe-A2.7B Model Under 2-bit Quantization

| Base Method | IDRE | BCOS | W2 |
|---|---|---|---|
| | ✗ | ✗ | 12.88 |
| GPTVQ | ✓ | ✗ | 11.03 |
| | ✗ | ✓ | 11.97 |
| | ✓ | ✓ | 9.43 |
| | ✗ | ✗ | 10.17 |
| VPTQ | ✓ | ✗ | 9.21 |
| | ✗ | ✓ | 9.77 |
| | ✓ | ✓ | 8.78 |

We conduct a speed evaluation in the decoder stage. In our tests with 1k input tokens, as shown in Table 6, the Qwen1.5-MoE-A2.7B model under 2-bit quantization (integrated with KBVQ

Table 6: Decoder speed test

| Model | Decoder Speed (Tokens/s) | | |
|---|---|---|---|
| | BF16 | Quantized | Speed Up |
| Qwen1.5-Moe-A2.7B | 22.31 | 35.24 | 1.58 |
| Qwen3-30B-A3B | 10.85 | 17.37 | 1.60 |

-MoE and corresponding VQ methods) achieves an inference speedup of over $1.5\times$, further validating the practical deployment value of the proposed framework in balancing accuracy retention and efficiency.

# 6 CONCLUSION

This paper proposes KBVQ-MoE, the first vector quantization framework tailored to Mixture-of-Experts (MoE) architectures. We address two critical challenges of applying vector quantization to MoE LLMs: (1) redundant representation among experts, which wastes limited codebook capacity, and (2) cumulative and amplified outputs bias, which manifests during expert aggregation and leads to severe distributional shifts. KBVQ-MoE integrates two key techniques: **Input-driven Redundancy Elimination (IDRE)**, which isolates shared structures and retains them at full precision, and **Bias-Corrected Output Stabilization (BCOS)**, which applies vector quantization only to expert-specific representations while correcting output distribution shifts through channel-wise affine compensation. Both theoretical analysis and empirical evaluations on diverse MoE LLMs (e.g., Qwen and Mixtral) demonstrate that KBVQ-MoE achieves superior performance under ultra-low-bit quantization, preserving accuracy with negligible runtime overhead. In future work, we will extend the framework to adaptive codebook design and expert pruning, further improving the deployment efficiency of large-scale MoE models in resource-constrained environments.

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

# A APPENDIX

## A.1 USE OF LLMS

Large language model (LLM) tools were employed solely for text polishing, including refinement of sentence structure, lexical optimization, and enhancement of language fluency.

All core components of this research—such as the conception of research ideas and framework, algorithm design and implementation, experimental protocol design, data collection and processing, execution and analysis of experiments, and validation of conclusions—were independently completed by the research team.

LLM tools were not involved in the conception of research content, generation of technical solutions, execution of experiments, or derivation of conclusions. The authors affirm full responsibility for the scientific validity, authenticity, and originality of the work in accordance with academic integrity standards.

## A.2 THEORETICAL PROOF OF THE INPUT COHERENCE BASIS

Let the linear mapping of the i-th expert be $y^{(i)} = W^{(i)}x$, with its quantized version denoted as $\tilde{W}^{(i)}$ and the error matrix defined as $E^{(i)} \triangleq \tilde{W}^{(i)} - W^{(i)}$. For the input covariance $C_X = \mathbb{E}[xx^T]$ that reflects the true distribution during inference, a natural definition of task distortion is the output mean squared error (MSE):

$$\mathcal{L} \triangleq \sum_{i=1}^{n} \mathbb{E}\left[\left\|\tilde{W}^{(i)}x - W^{(i)}x\right\|_2^2\right] = \sum_{i=1}^{n} \text{Tr}\left(E^{(i)}C_X E^{(i)T}\right). \tag{9}$$

This metric directly measures the impact of quantization on the output side, rather than the unweighted difference of the weights themselves. Perform the Karhunen–Loève Transform (KLT) on $C_X$, such that $C_X = U_{\text{KLT}}\Lambda_{\text{KLT}}U_{\text{KLT}}^T$, where $\Lambda_{\text{KLT}} = \text{diag}(\lambda_1 \geq \cdots \geq \lambda_{ic})$. Define the input coherence basis as $U_X \triangleq U_{\text{KLT}}\Lambda_{\text{KLT}}^{1/2}$. Then the following identity holds:

$$\text{Tr}(EC_X E^T) = \text{Tr}\left(EU_X(U_X)^T E^T\right) = \|EU_X\|_F^2, \tag{10}$$

thus

$$\mathcal{L} = \sum_{i=1}^{n} \left\|(\tilde{W}^{(i)} - W^{(i)})U_X\right\|_F^2. \tag{11}$$

This indicates that: in the input coherence space, the output MSE is equivalent to the Frobenius norm of the weight error. Therefore, all "extraction/quantization" operations aimed at reducing output distortion should be performed in this space. Denote $\hat{W}^{(i)} \triangleq W^{(i)}U_X$.

## A.3 SPECTRAL CHARACTERIZATION OF THE OPTIMAL SHARED SUBSPACE (KLT–SVD).

To formalize the redundancy removal performed by IDRE, we stack all expert weights along the output dimension:

$$W \triangleq \begin{bmatrix} \hat{W}^{(1)} \\ \vdots \\ \hat{W}^{(n)} \end{bmatrix} \in \mathbb{R}^{(no_c)\times i_c}, \qquad S \triangleq W^\top W. \tag{12}$$

Here $\hat{W}^{(i)}$ denotes the $i$-th expert weight expressed in the input KLT coordinates introduced in Appendix. A.2. The matrix $S$ is the (input–aligned) Gram matrix of all experts. In practice, IDRE computes a truncated SVD of the stacked matrix $W \in \mathbb{R}^{(n \cdot o_c)\times i_c}$: $W = U\Sigma V^\top$. Note that the Gram matrix $S = W^\top W$ admits the eigen-decomposition $S = V\Sigma^2 V^\top$, i.e., the eigenvalues of $S$ are the squared singular values of $W$, and the eigenvectors of $S$ coincide with the right singular vectors of $W$. Therefore, selecting the top–$k$ eigenvectors of $S$ is exactly equivalent to performing a rank–$k$ truncated SVD on $W$ and keeping the top–$k$ right singular vectors. The parameter $k$ used in our analysis and implementation is the dimension of this shared right-singular subspace.

**Optimal shared subspace.** We seek a $k$-dimensional shared subspace that maximizes the stacked energy of all experts:

$$\max_{U \in \mathbb{R}^{i_c \times k}, \, U^\top U = I} \mathrm{Tr}(U^\top S U) \;=\; \max_{U^\top U = I} \left\| W U \right\|_F^2. \tag{13}$$

By Ky Fan's theorem, the optimizer $U_k$ is given by the top–$k$ eigenvectors of $S$ associated with its largest eigenvalues $\{\sigma_j^2\}_{j=1}^k$. Let $P_k \triangleq U_k U_k^\top$ be the projector onto this shared subspace. In the KLT space, each expert admits the orthogonal decomposition

$$\check{W}^{(i)} \;=\; \underbrace{\check{W}^{(i)} P_k}_{\text{shared (redundancy)}} \;+\; \underbrace{\check{W}^{(i)} (I - P_k)}_{\text{specific (difference)}}, \tag{14}$$

and mapping back to the original coordinates yields $W_{\text{share}}^{(i)} = \hat{W}^{(i)} P_k T^{-1}$ and $W_{\text{spec}}^{(i)} = W^{(i)} - W_{\text{share}}^{(i)}$, where $T$ is the KLT transform defined in Sec. A.2. Intuitively, the shared component captures directions that are simultaneously important for many experts, while the specific component collects expert–unique variations.

**Why KLT before SVD.** Let $x$ be the input to the MoE layer with covariance $\Sigma_X = \mathbb{E}[xx^\top] = U_X \Lambda_X U_X^\top$ and define the input KLT transform by $T = U_X \Lambda_X^{1/2}$. In this basis, the stacked Gram matrix takes the form

$$S = W^\top W = T^\top \Big( \sum_{i=1}^n W^{(i)\top} W^{(i)} \Big) T = \Lambda_X^{1/2} U_X^\top \Big( \sum_{i=1}^n W^{(i)\top} W^{(i)} \Big) U_X \Lambda_X^{1/2}. \tag{15}$$

Thus the spectrum $\{\sigma_j^2\}$ of $S$ jointly reflects *both* the input energy (through $\Lambda_X$) and the across–expert weight energy (through $\sum_i W^{(i)\top} W^{(i)}$) along each input principal direction. Selecting the top–$k$ eigenvectors of $S$ therefore retains those directions that are simultaneously dominant in the input distribution and heavily utilized across experts, which is precisely the notion of "shared" structure we aim to preserve in full precision.

**Truncation error and redundancy removal.** The residual (expert–specific) part after projecting onto $P_k$ satisfies

$$\sum_{i=1}^n \left\| \check{W}^{(i)} (I - P_k) \right\|_F^2 \;=\; \left\| W (I - P_k) \right\|_F^2 \;=\; \sum_{j > k} \sigma_j^2, \tag{16}$$

where $\{\sigma_j^2\}_{j=1}^{i_c}$ are the eigenvalues of $S$ sorted in decreasing order. Hence the truncation error is *exactly* controlled by the tail eigenvalues discarded by $P_k$. Correspondingly, the redundancy removal ratio of IDRE at rank $k$ can be quantified as

$$\rho_k \;\triangleq\; \frac{\sum_{j=1}^k \sigma_j^2}{\sum_{j=1}^{i_c} \sigma_j^2}, \tag{17}$$

which measures the fraction of stacked expert energy captured by the shared subspace.

This characterization provides an explicit trade-off: increasing $k$ monotonically raises $\rho_k$ and reduces the truncation error in equation 16, but also enlarges the full-precision shared component. Empirically, for MoE LLMs such as Qwen1.5-MoE-A2.7B and Qwen3-30B-A3B, we observe that the spectrum of $S$ is strongly low-rank and decays rapidly (approximately heavy–tailed). As shown in the Fig.5, it demonstrates the obvious low-rank characteristics of the expert groups and their posterior singularity.. Under a simple power-law approximation $\sigma_j^2 \propto j^{-\alpha}$ with $\alpha > 1$, the residual fraction $1 - \rho_k$ behaves like $Ck^{1-\alpha}$; once $k$ exceeds a moderate threshold, the marginal gain from further increasing $k$ quickly diminishes. Across layers and models, choosing $k \approx i_c/128$ typically yields $\rho_k \approx 0.6$–$0.8$, while larger ranks (e.g., $k = i_c/64$) provide only marginal perplexity improvements at noticeably higher storage cost.Tab.7 shows a comparison of k selection for two moe models of Qwen, and the low rank of 1/128 can already achieve very good results. This explains why $k = i_c/128$ serves as a robust operating point that balances reconstruction error (controlled by the tail eigenvalues) and the overhead of the shared full-precision subspace across different models and tasks.

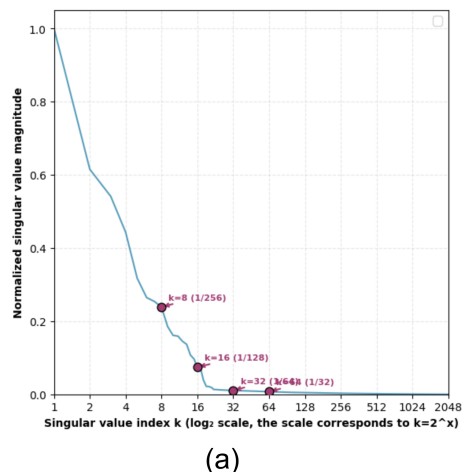 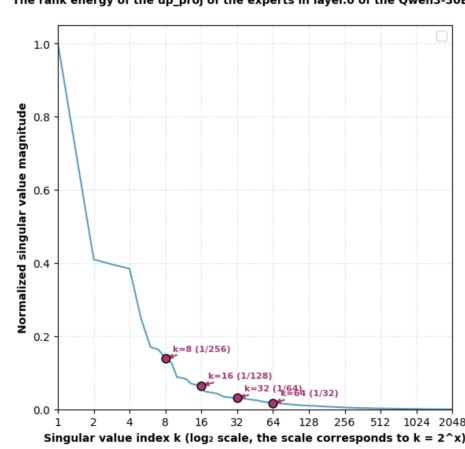

Figure 5: The low-rank characteristics after expert fusion activation of the first block, (a) Qwen1.5-MoE-A2.7B, (b) Qwen3-30B-A3B

Table 7: Low-rank value ablation

| k | 1/256 | 1/128 | 1/64 | 1/48 | 1/32 |
|---|---|---|---|---|---|
| Compress Ratio | 2.05 | 2.08 | 2.11 | 2.15 | 2.2 |
| Qwen1.5-MoE-A2.7B | 20.71 | 9.61 | 9.60 | 9.58 | 9.58 |
| Qwen3-30B-A3B | 15.30 | 11.87 | 11.30 | 11.30 | 11.01 |

### A.4 SYSTEM BIAS CORRECTION

In the paper, we reformulate bias correction as a residual affine transform on the quantized output:

$$y_{\text{corr}} = W_{\text{VQ}}x + s \odot (W_{\text{VQ}}x) + b = (1 + s) \odot (W_{\text{VQ}}x) + b, \tag{18}$$

where $s, b \in \mathbb{R}^{oc}$ denote channel-wise residual scaling and bias, respectively. We now show that this correction is statistically optimal under the mean squared error (MSE) criterion.

**1. Problem Setup**  Let the full-precision output be $y = Wx$ and the quantized output be $\hat{y} = W_{\text{VQ}}x$. Our goal is to determine $s$ and $b$ such that

$$y_{\text{corr}} = (1 + s) \odot \hat{y} + b$$

approximates $y$ as closely as possible. The optimization problem can be written as

$$(s^*, b^*) = \arg\min_{s,b} \; \mathbb{E}\big[\|y - ((1 + s) \odot \hat{y} + b)\|^2\big]. \tag{19}$$

**2. Closed-form Solution**  For the $j$-th output channel, this reduces to:

$$(s_j^*, b_j^*) = \arg\min_{s_j,b_j} \; \mathbb{E}\big[(y_j - ((1 + s_j)\hat{y}_j + b_j))^2\big]. \tag{20}$$

This is equivalent to a univariate linear regression with slope $\alpha_j = 1 + s_j$ and intercept $b_j$. Its closed-form solution is

$$\alpha_j^* = \frac{\text{Cov}(y_j, \hat{y}_j)}{\text{Var}(\hat{y}_j)}, \qquad b_j^* = \mu_{y_j} - \alpha_j^* \mu_{\hat{y}_j}, \tag{21}$$

and therefore

$$s_j^* = \alpha_j^* - 1. \tag{22}$$

**3. Approximation under High Correlation**  Because $y_j$ and $\hat{y}_j$ differ only by quantization noise, they are highly correlated. Thus, we approximate

$$\mathrm{Cov}(y_j, \hat{y}_j) \approx \sigma_{y_j}\, \sigma_{\hat{y}_j}. \tag{23}$$

Substituting into the regression solution gives

$$s_j^* \approx \frac{\sigma_{y_j}}{\sigma_{\hat{y}_j}} - 1, \qquad b_j^* = \mu_{y_j} - (1 + s_j^*)\mu_{\hat{y}_j}. \tag{24}$$

**4. Conclusion**  This yields exactly the mean–variance matching rule used in the main text, showing that the residual affine correction is not a heuristic adjustment but an *unbiased MMSE-optimal estimator*.

### A.5 COMPRESSION CONDITION

In the current experiment, all methods are matched by weight bit-width (2 bits or 3 bits), while the additional overhead of KBVQ-MoE (shared low-rank components and per-channel biases) only slightly increases the average number of bits per weight (by approximately 0.08 bits). In the table8, we report the actual storage usage and effective bits per parameter of all baselines, and provide comparisons under equal compression ratios.

Table 8: compression achieved by various techniques

| Bit | Method | Compress ratio | Qwen1.5-MoE-A2.7B | Qwen3-30B-A3B | Mixtral-8x7B | DeepseekV2-Lite |
|------|----------|------|------|------|------|------|
| 16 | FP16 | 0% | 27.9GB | 59.1GB | 89.6GB | 29.3GB |
| 2 | RTN | 87.5% | 4.3GB | 8.4GB | 12.6GB | 4.3GB |
| 2.25 | GPTQ | 85.9% | 4.7GB | 9.3GB | 13.5GB | 4.7GB |
| 2 | MoeQuant | 87.5% | 4.3GB | 8.4GB | 12.6GB | 4.3GB |
| 2.08 | VQ | 87% | 4.3GB | 8.7GB | 13GB | 4.4GB |
| 2.08 | KBVQ-MoE | 87% | 4.3GB | 8.7GB | 13GB | 4.4GB |

We provide a comprehensive quantification from **three perspectives** : offline computation overhead, inference-time overhead, and memory/parameter cost. These analyses complement the "Decoder speed test" in Table 6 in manuscript and present a complete view of the efficiency characteristics of KBVQ-MoE.

(1) Offline Computation Overhead (Does Not Affect Inference Latency)

KBVQ-MoE introduces an additional one-time KLT-guided SVD decomposition during the calibration phase. This operation is applied after stacking all expert weights: The cost of KLT-SVD is comparable to a single forward pass on the calibration set. It occurs entirely offline during calibration. It introduces no additional inference-time latency or compute cost. Therefore, the additional computation introduced by KBVQ-MoE does not affect actual inference performance.

(2) Inference-Time Overhead (Nearly Negligible)

During inference, KBVQ-MoE introduces only: Two extra parameters per output channel (scale and bias). A lightweight per-channel affine correction after each expert's linear layer: $y_{\mathrm{corr}} = (1 + s) \odot (W_{\mathrm{VQ}}x) + b$. This involves only element-wise additions and multiplications, whose complexity is far lower than the original MatVec operation inside each expert MLP. Empirically, this affine correction accounts for less than 0.1% of the expert forward FLOPs , consistent with the 1.5–1.6× inference speedup reported in Table 6. Thus, KBVQ-MoE introduces minimal additional compute overhead during inference.

(3) Detailed Memory and Parameter Overhead Analysis

Let: Expert weight dimension: $m \times l$, Number of experts: n, IDRE rank ratio: k, VQ subvector length: v, Quantization bit-width: b.

The total bit consumption of KBVQ-MoE consists of four parts:  Original FP16 parameters:$16nml$, IDRE shared low-rank components:$16(m + ln)\min(m,l)k$, VQ codebook + indices:$mlbn + 16 \cdot 2^{bv} \cdot v \cdot n$, BCOS scale/bias (per channel):$16 \cdot 2 \cdot ln$, Total compression ratio:$\frac{16(m+ln)\min(m,l)k+mlbn+2^{(bv+4)}vn+32ln}{16nml}$.

Example: Qwen1.5-MoE-2.7B (gate_proj weight) , Weight dimension: $5632 \times 2048$, Number of experts: 64, Original storage: $5632 \times 2048 \times 64 \times 16$; bit $= 1.38$; GB, KBVQ-MoE configuration: 2-bit VQ, vector length $= 4$, $k = \frac{1}{128}$, Total compressed storage: $((5632 + 2048 \times 64) \cdot 2048/128 \cdot 16) + (5632 \cdot 2048 \cdot 2 \cdot 64 + 2^8 \cdot 4 \cdot 16 \cdot 64) + (2048 \cdot 2 \cdot 16 \cdot 64)$, bit $= 0.18$, GB.

This corresponds to an 87% compression ratio , equivalent to 2.08 effective bits per weight.

Through the analyses above, we provide a complete efficiency characterization of KBVQ-MoE across: Offline computation: acceptable one-time cost, no inference impact. Inference computation: negligible additional overhead (<0.1% FLOPs), Memory cost: clear and formula-based quantification (87% compression demonstrated).

## A.6 4BIT ABLATION EXPERIMENT

Our primary focus in this work is to address the challenges of ultra–low-bit compression (e.g., 2–3 bits), where existing methods often suffer from severe accuracy degradation. In higher-bit regimes, standard scalar quantization (SQ) can already achieve strong performance, and the relative advantage of vector quantization (VQ) gradually diminishes. Moreover, our method is based on vector quantization, whose codebook training via $k$-means clustering incurs a computational cost that grows exponentially with the target bit-width, making VQ less attractive in practice for high-bit settings. For completeness, we additionally report the performance of KBVQ-MoE at 4-bit in Tab.9, which shows that our method remains competitive in this regime, even though ultra–low-bit compression is the main target of our design.

Table 9: 4bit ablation experiment

| Model | Method | W2 | ARC-C | ARC-E | PIQA | WI |
|---|---|---|---|---|---|---|
| Qwen1.5-MoE-A2.7B | FP16 | 7.22 | 69.53 | 44.03 | 80.47 | 69.30 |
| | RTN | 10.83 | 64.95 | 41.12 | 76.84 | 67.28 |
| | GPTQ | 7.43 | 68.03 | 43.78 | 78.90 | 68.14 |
| | MoeQuant | 7.55 | 68.97 | 44.04 | 79.76 | 67.86 |
| | VQ | 8.92 | 64.68 | 40.88 | 77.20 | 64.91 |
| | KBVQ-MoE | 7.59 | 69.05 | 44.39 | 80.33 | 68.56 |
| Mixtral-8x7B | FP16 | 3.84 | 85.39 | 66.38 | 85.20 | 76.72 |
| | RTN | 5.41 | 75.84 | 55.48 | 77.84 | 70.73 |
| | GPTQ | 4.03 | 84.89 | 63.78 | 84.21 | 74.97 |
| | MoeQuant | 4.12 | 83.88 | 63.70 | 84.20 | 75.02 |
| | VQ | 4.27 | 84.18 | 64.21 | 84.93 | 74.81 |
| | KBVQ-MoE | 4.16 | 84.97 | 65.04 | 84.84 | 74.17 |

## A.7 ALL EXPERIMENTS

Table 10: Comparison experiment of the Qwen1.5-MoE-A2.7B model

| Bit | Method | W2 | ARC-E | ARC-C | HE | LAMBADA-O | LAMBADA-S | PIQA | WI | AVG |
|---|---|---|---|---|---|---|---|---|---|---|
| 16 | FP16 | 7.22 | 69.53 | 44.03 | 77.26 | 71.28 | 64.62 | 80.47 | 69.30 | 68.07 |
| 2 | RTN | 638509 | 24.87 | 27.90 | 26.39 | 0 | 0 | 51.47 | 48.86 | 25.64 |
| | GPTQ | 12.69 | 47.14 | 30.89 | 60.77 | 43.72 | 34.81 | 69.97 | 56.20 | 49.07 |
| | MoeQuant | 583542 | 34.54 | 35.09 | 35.55 | 12.03 | 8.05 | 59.08 | 58.21 | 34.64 |
| | VQ | 26.95 | 58.16 | 34.64 | 63.35 | 28.90 | 22.92 | 71.82 | 55.09 | 47.84 |
| | KBVQ-MoE | 9.61 | 65.66 | 39.59 | 64.52 | 67.16 | 60.20 | 77.20 | 65.11 | 62.78 |
| 3 | RTN | 116689 | 25.84 | 24.66 | 27.52 | 0.14 | 0.02 | 51.69 | 49.88 | 25.68 |
| | GPTQ | 7.58 | 68.60 | 43.34 | 75.35 | 68.68 | 62.80 | 79.22 | 66.54 | 66.36 |
| | MoeQuant | 7.87 | 68.56 | 44.21 | 74.29 | 70.54 | 64.97 | 79.51 | 65.45 | 66.79 |
| | VQ | 11.47 | 60.86 | 36.18 | 60.56 | 54.67 | 41.30 | 75.57 | 62.43 | 55.94 |
| | KBVQ-MoE | 7.74 | 68.90 | 44.54 | 75.08 | 72.81 | 66.64 | 80.14 | 67.80 | 67.99 |

Table 11: Comparison experiment of the Qwen3-30B-A3B model

| Bit | Method | W2 | ARC-E | ARC-C | HE | LAMBADA-O | LAMBADA-S | PIQA | WI | AVG |
|---|---|---|---|---|---|---|---|---|---|---|
| 16 | FP16 | 8.70 | 79.25 | 56.40 | 79.60 | 64.70 | 62.91 | 80.30 | 70.48 | 70.24 |
| 2 | RTN | 765922 | 24.49 | 26.71 | 25.93 | 0 | 0 | 51.31 | 52.80 | 25.89 |
| | GPTQ | 438.42 | 34.49 | 27.85 | 33.89 | 14.62 | 18.11 | 52.23 | 50.23 | 33.06 |
| | MoeQuant | 37465 | 29.81 | 26.59 | 25.38 | 7.93 | 11.78 | 50.74 | 50.41 | 28.94 |
| | VQ | 115.30 | 33.08 | 24.49 | 35.74 | 6.79 | 7.35 | 56.86 | 49.96 | 30.61 |
| | KBVQ-MoE | 11.87 | 70.33 | 47.61 | 67.49 | 60.68 | 54.34 | 76.66 | 66.46 | 63.37 |
| 3 | RTN | 68657 | 27.48 | 24.91 | 27.38 | 0.02 | 0.21 | 51.80 | 49.09 | 25.84 |
| | GPTQ | 11.32 | 75.92 | 50.29 | 71.32 | 59.83 | 53.93 | 69.12 | 64.19 | 63.51 |
| | MoeQuant | 24.85 | 30.49 | 37.77 | 31.49 | 18.38 | 35.11 | 55.82 | 52.23 | 37.32 |
| | VQ | 18.72 | 57.83 | 40.87 | 63.23 | 40.44 | 32.51 | 71.82 | 57.70 | 52.06 |
| | KBVQ-MoE | 9.26 | 77.27 | 53.24 | 75.53 | 66.37 | 62.35 | 78.89 | 70.01 | 69.09 |

Table 12: Comparison experiment of the Mixtral-8x7B model

| Bit | Method | W2 | ARC-E | ARC-C | HE | LAMBADA-O | LAMBADA-S | PIQA | WI | AVG |
|---|---|---|---|---|---|---|---|---|---|---|
| 16 | FP16 | 3.88 | 85.39 | 66.38 | 85.95 | 77.28 | 73.06 | 85.20 | 76.72 | 78.57 |
| 2 | RTN | 274952 | 25.38 | 28.41 | 26.09 | 0 | 0 | 50.76 | 46.25 | 25.27 |
| | GPTQ | 5.69 | 72.35 | 48.89 | 76.95 | 68.39 | 61.44 | 77.15 | 67.72 | 67.56 |
| | MoeQuant | 13.43 | 49.85 | 38.93 | 40.12 | 22.98 | 18.94 | 60.38 | 49.91 | 40.16 |
| | VQ | 5.99 | 72.73 | 44.62 | 70.08 | 50.46 | 36.70 | 74.42 | 65.51 | 59.22 |
| | KBVQ-MoE | 5.39 | 81.52 | 58.19 | 80.62 | 80.57 | 72.48 | 82.59 | 73.88 | 75.69 |
| 3 | RTN | 45136 | 26.39 | 28.50 | 25.58 | 0 | 0 | 51.58 | 48.54 | 25.80 |
| | GPTQ | 4.17 | 84.01 | 64.42 | 85.12 | 76.77 | 71.76 | 83.79 | 76.16 | 77.43 |
| | MoeQuant | 5.45 | 80.23 | 57.39 | 80.91 | 71.18 | 64.95 | 78.91 | 71.91 | 72.21 |
| | VQ | 5.52 | 81.65 | 58.45 | 82.12 | 72.97 | 67.44 | 81.88 | 73.48 | 73.98 |
| | KBVQ-MoE | 4.07 | 84.89 | 63.05 | 84.99 | 80.50 | 75.53 | 83.90 | 75.61 | 78.35 |

Table 13: Comparison experiment of the DeepSeekV2-Lite model

| Bit | Method | W2 | ARC-E | ARC-C | HE | LAMBADA-O | LAMBADA-S | PIQA | WI | AVG |
|---|---|---|---|---|---|---|---|---|---|---|
| 16 | FP16 | 5.92 | 76.22 | 48.98 | 77.91 | 72.33 | 67.90 | 80.20 | 71.19 | 70.68 |
| 2 | RTN | 174653 | 24.12 | 26.45 | 26.30 | 0 | 0 | 49.89 | 49.09 | 25.12 |
| | GPTQ | 8.49 | 63.47 | 37.63 | 65.45 | 52.53 | 48.55 | 74.59 | 64.09 | 58.04 |
| | MoeQuant | 25893 | 26.94 | 25.40 | 25.45 | 0 | 0 | 50.14 | 51.29 | 25.59 |
| | VQ | 10.96 | 53.99 | 38.71 | 50.92 | 48.19 | 40.11 | 57.73 | 59.33 | 49.85 |
| | KBVQ-MoE | 7.94 | 68.47 | 40.60 | 67.93 | 64.91 | 55.94 | 75.91 | 67.91 | 63.10 |
| 3 | RTN | 97.75 | 43.64 | 27.05 | 42.56 | 2.68 | 1.84 | 66.00 | 49.49 | 33.32 |
| | GPTQ | 6.98 | 74.04 | 46.35 | 76.44 | 70.4 | 65.65 | 79.05 | 70.27 | 68.89 |
| | MoeQuant | 7.52 | 70.25 | 42.91 | 71.19 | 70.04 | 65.32 | 76.84 | 67.83 | 66.34 |
| | VQ | 7.94 | 67.91 | 43.22 | 68.81 | 67.90 | 60.30 | 67.82 | 59.83 | 62.26 |
| | KBVQ-MoE | 6.95 | 73.74 | 45.73 | 74.36 | 72.62 | 66.33 | 79.05 | 69.30 | 68.73 |

A.8  KBVQ-MoE ALGORITHM

---

**Algorithm 1** KBVQ-MoE: KLT-guided SVD with Bias-Corrected VQ for MoE Models

---

**Input:** Expert weight matrices $\{W^{(i)}\}_{i=1}^n$, input activations $X$, codebook size $K$, sub-vector length $d$

**Output:** Quantized weights $\{\hat{W}^{(i)}\}_{i=1}^n$ with bias correction parameters $(s, b)$

    **Pre-Process: KLT-guided SVD (Redundancy Removal)**

1: Compute input covariance: $C_X = \frac{1}{B-1} X^\top X \in \mathbb{R}^{ic \times ic}$
2: Eigen-decompose: $C_X = U_{\text{KLT}}^\top \Lambda_{\text{KLT}} U_{\text{KLT}}$
3: Input-coherent basis: $U_X = U_{\text{KLT}} \Lambda_{\text{KLT}}^{1/2}$;
4: **for** $i = 1$ **to** $n$ **do**
5:     Project expert weights (right projection): $\widetilde{W}^{(i)} = W^{(i)} U_X$
6: **end for**
7: Stack all experts (row-wise): $\bar{W} = [\widetilde{W}^{(1)}; \cdots ; \widetilde{W}^{(n)}] \in \mathbb{R}^{(n*oc) \times ic}$
8: Product SVD: $\bar{W} = (U \Sigma V^\top)^T$          $\triangleright V \in \mathbb{R}^{(oc) \times n*oc}, U \in \mathbb{R}^{ic \times ic}$
9: Select top-$k$: $U_k = U_{[:,1:k]}, V_k = V_{[:,1:k]}, \Sigma_k = \Sigma_{1:k,1:k}$
10: Partition $V_k$ by experts: $V_k = [\Sigma_k V_k^{(1)}; \ldots ; \Sigma_k V_k^{(n)}]$, each $V_k^{(i)} \in \mathbb{R}^{oc \times k}$
11: **for** $i = 1$ **to** $n$ **do**
12:     Define shared output loader: $U_{\text{share}} \leftarrow U_X^{-1} U_k$          $\triangleright ic \times k$
13:     Shared part in projected basis: $\widehat{W}_{\text{share}}^{(i)} = (U_{\text{share}} (V_k^{(i)})^T)^T$      $\triangleright oc \times ic$
14:     Map back to original input space: $W_{\text{share}}^{(i)} = \widehat{W}_{\text{share}}^{(i)}$
15:     Special (residual) part: $W_{\text{quant}}^{(i)} = W^{(i)} - W_{\text{share}}^{(i)}$
16: **end for**

    **Quantization: Vector Quantization of Special Part**

17: **for** $i = 1$ to $n$ **do**
18:     Split $W_{\text{quant}}^{(i)}$ into sub-vectors $\{z\}$
19:     Initialize codebook via K-means++
20:     Train VQ codebook $C = \{c_1, \ldots, c_K\}$ by k-means
21:     **for** each sub-vector $z$ **do**
22:         Assign index: $q = \arg\min_j \|z - c_j\|^2$
23:         Replace: $z_q = c_q$
24:     **end for**
25:     Form quantized special part: $W_{\text{quant,VQ}}^{(i)}$
26: **end for**

    **Post-Process: Bias Correction**

27: **for** $i = 1$ **to** $n$ **do**
28:     Define quantized weights: $\hat{W}^{(i)} = W_{\text{share}}^{(i)} + W_{\text{quant,VQ}}^{(i)}$
29:     Estimate per-channel statistics from calibration data:
        $\mu_y, \sigma_y$ for original outputs $y = W^{(i)} x$
        $\mu_{\hat{y}}, \sigma_{\hat{y}}$ for quantized outputs $\hat{y} = \hat{W}^{(i)} x$
30:     Compute correction parameters:
        $s_j = \frac{\sigma_{y_j}}{\sigma_{\hat{y}_j}} - 1, \quad b_j = \mu_{y_j} - (1 + s_j) \mu_{\hat{y}_j}$
31:     Corrected output: $y_{\text{corr}} = (1 + s) \odot (\hat{W}^{(i)} x) + b$
32: **end for**
33: **return** $U_{\text{share}}, V_k, C, (s, b)$

---

A.9  ABLATION OF IDRE AND BCOS

To further clarify the contribution of each component within the IDRE and BCOS modules, we conduct fine-grained, component-wise ablations at 2-bit on Qwen1.5-MoE-A2.7B. This directly

addresses the reviewer's concern regarding the lack of quantitative analysis for the internal steps of each module.

(a) Effect of the KLT step in IDRE. IDRE consists of (i) an input-driven KLT transform and (ii) a subsequent SVD on the stacked expert weights in the KLT space. To isolate the effect of the KLT step, we compare the full IDRE against a variant that performs SVD directly on the original stacked weights (without KLT alignment). The results at 2-bit are summarized in Tab.14.

Table 14: Ablation of KLT in IDRE on Qwen1.5-MoE-A2.7B (2-bit).

| Model | IDRE | W2 | HE | WI |
|-------|------|-----|------|------|
| Qwen1.5-MoE-A2.7B | w/o KLT | 11.77 | 35.89 | 63.01 |
| | KLT | **9.61** | **39.59** | **65.11** |

(b) Effect of mean and variance correction in BCOS. BCOS performs channel-wise bias correction using both the mean and variance of the channel outputs. To disentangle their respective roles, we evaluate four variants: (i) no correction, (ii) mean-only, (iii) variance-only, and (iv) full mean+variance correction. The 2-bit results on Qwen1.5-MoE-A2.7B are given in Tab.15.

Table 15: Ablation of mean and variance correction in BCOS on Qwen1.5-MoE-A2.7B (2-bit).

| Model | Mean | Variance | W2 | HE | WI |
|-------|------|----------|-----|------|------|
| Qwen1.5-MoE-A2.7B | ✗ | ✗ | 11.03 | 35.18 | 63.49 |
| | ✓ | ✗ | 11.01 | 35.35 | 63.40 |
| | ✗ | ✓ | 10.38 | 36.70 | 64.23 |
| | ✓ | ✓ | **9.61** | **39.59** | **65.11** |

### A.10 COMPARISON OF MoE COMPRESSION METHODS

The core designs of D2-MoE, SubMoE, and EAC-MoE all revolve around "expert layer structure optimization", which belongs to a different technical branch of MoE compression compared to KBVQ-MoE's "weight quantization". The specific differences can be summarized as follows:

- SubMoE: Merges similar experts through subspace clustering and extracts cross-expert shared low-rank features via SVD to reduce the number of experts. Essentially, it is a structural compression method combining "expert-level pruning + low-rank decomposition". The compression ratio is mainly determined by the expert merging ratio (e.g., 50% means merging half of the experts).

- D2-MoE: Decomposes expert weights into a "shared base matrix + expert-specific delta matrix" and performs low-rank approximation on the delta matrix to reduce the number of parameters. Essentially, it is a structural compression method combining "weight decomposition + low-rank approximation". The compression ratio is determined by the rank truncation ratio of the delta matrix (e.g., retaining 60% of the rank).

- EAC-MoE: Starts from "expert selection behavior" and jointly models two operations: quantization and pruning. It proposes QESC (Quantization with Expert-Selection Calibration), which explicitly calibrates the router during low-bit quantization to mitigate expert selection bias caused by quantization noise. It also proposes PESF (Pruning based on Expert-Selection Frequency), which prunes "cold experts" using their actual selection frequency to further improve inference speed and reduce memory usage while minimizing accuracy loss. Thus, EAC-MoE can be regarded as a "routing-aware quantization + expert pruning hybrid compression scheme".

- KBVQ-MoE (ours): Does not change the number or structure of experts. First, it extracts cross-expert shared low-rank subspaces in the input KLT space via IDRE to explicitly remove redundancy, then performs vector quantization on expert-specific residuals. Essentially, it is an "MoE structure-aware weight quantization method". The compression ratio is determined

by the quantization bit-width (2–3 bits) and codebook reuse efficiency, achieving 80%–90% memory compression under the same structural constraints.

Overall, D2-MoE/SubMoE are more inclined to directly reduce the number of parameters (removing experts / reducing rank), EAC-MoE follows a hybrid route of structural pruning + quantization, while KBVQ-MoE focuses on achieving high compression ratios through structure-aware low-bit quantization while preserving the original expert structure. Different routes may exhibit distinct trade-offs in terms of "expert specificity preservation", "inference speed", and "implementation complexity", which need to be characterized through empirical comparisons.

**Fair Comparison Experiments: Settings and Results.** To ensure fairness, we uniformly use Mixtral-8×7B as the baseline model (retaining the original expert structure: 8 experts, 32 layers) and adopt identical experimental settings:

- Evaluation tasks: WikiText2 (language modeling, metric: perplexity (PPL), lower is better), ARC-Challenge (ARC_C), ARC-Easy (ARC_E), WinoGrande (WinG) (all zero-shot inference, metric: accuracy, higher is better);

- Compression ratio definition: Memory saving is uniformly calculated as "the ratio of compressed parameter storage to the original FP16 model" to avoid misunderstandings caused by inconsistent definitions of "compression ratio" across different methods;

- Experimental environment: NVIDIA RTX A100 GPU, PyTorch 2.1, evaluation tool: LM-Evaluation-Harness (v0.4.0).

The comparison results are shown in the Tab.16 (Note: ARC_C, ARC_E, and WinG values are accuracy, retained to two decimal places for precision):

Table 16: Ablation KBVQ—MoE and other MoE compression methods

| Method | Memory saving | WikiText2 (PPL) | ARC_C | ARC_E | WinG |
|---|---|---|---|---|---|
| Sub-MoE | 50% | 6.97 | 0.45 | 0.75 | 0.72 |
| D2-MoE | 60% | 6.46 | 0.38 | 0.72 | 0.71 |
| EAC-MoE | 84% | 4.58 | 0.55 | 0.81 | 0.75 |
| KBVQ-MoE (ours) | 87% | 4.07 | 0.63 | 0.85 | 0.76 |

In the experiments on Mixtral-8×7B, significant differences exist among the three compression approaches: KBVQ-MoE (quantization compression) achieves an 87% memory saving, which is significantly higher than that of Sub-MoE (50%) and D2-MoE (60%), and slightly higher than that of EAC-MoE (84%). Moreover, it performs optimally across all tasks: its Perplexity (PPL) on WikiText2 is 2.39-2.90 lower than that of Sub-MoE/D2-MoE and 0.51 lower than that of EAC-MoE; its accuracy on ARC_C is 0.18-0.25 higher than that of Sub-MoE/D2-MoE and 0.08 higher than that of EAC-MoE. These results confirm that KBVQ-MoE can well retain the expressive ability of MoE experts under a high compression ratio (>80%).

EAC-MoE (hybrid compression), which combines routing calibration and expert pruning based on selection frequency, outperforms Sub-MoE and D2-MoE (structural compression) in performance. However, KBVQ-MoE still shows better performance under a similar compression ratio and does not require modifying the expert structure. Additionally, KBVQ-MoE preserves the original expert topology, achieving an accuracy of 0.63 on complex reasoning tasks (e.g., ARC_C)—far exceeding the 0.38-0.45 of Sub-MoE/D2-MoE and the 0.55 of EAC-MoE. It also features greater flexibility in deployment and migration: it can be directly adapted to various MoE LLMs (e.g., Qwen, DeepSeek) with only one round of offline calibration and codebook training, demonstrating prominent comprehensive advantages.

## A.11 CHALLENGING BENCHMARKS

We carried out ablation studies between our method and diverse baselines on more arduous tasks shown on Tab.17. Main tasks include: MMLU(Wang et al., 2024), MathQA(Amini et al., 2019), GSM8K(Cobbe et al., 2021), HumanEval(Chen et al., 2021).

Table 17: more challenging benchmarks

| Model | Bits | Method | MMLU | MathQA | GSM8K | HumanEval |
|---|---|---|---|---|---|---|
| Qwen1.5-MoE-A2.7B | 16 | FP16 | 59.60 | 37.55 | 62.55 | 32.32 |
| | 2 | GPTQ | 26.94 | 19.33 | 15.42 | 13.88 |
| | | MoeQuant | 34.75 | 22.42 | 23.21 | 18.12 |
| | | VQ | 48.79 | 28.34 | 46.51 | 28.03 |
| | | KBVQ-MoE | 52.16 | 30.75 | 56.39 | 30.11 |
| Qwen3-30B-A3B | 16 | FP16 | 79.5 | 58.53 | 85.44 | 71.5 |
| | 2 | GPTQ | 42.11 | 34.9 | 49.34 | 50.12 |
| | | MoeQuant | 45.1 | 39.86 | 57.38 | 55.32 |
| | | VQ | 60.94 | 49.87 | 64.83 | 61.2 |
| | | KBVQ-MoE | 68.19 | 50.11 | 78.92 | 66.89 |
| DeepSeekMoE-16B | 16 | FP16 | 44.60 | 31.49 | 20.16 | 26.83 |
| | 2 | GPTQ | 28.48 | 18.82 | 11.26 | 14.87 |
| | | MoeQuant | 34.82 | 24.93 | 12.88 | 16.95 |
| | | VQ | 30.78 | 24.49 | 13.55 | 17.89 |
| | | KBVQ-MoE | 41.39 | 28.97 | 17.32 | 22.62 |
| Mixtral-7x8B | 16 | FP16 | 70.50 | 42.41 | 65.88 | 32.93 |
| | 2 | GPTQ | 40.80 | 22.96 | 5.89 | 9.97 |
| | | MoeQuant | 49.39 | 27.11 | 20.67 | 14.74 |
| | | VQ | 55.84 | 30.83 | 49.4 | 23.21 |
| | | KBVQ-MoE | 61.11 | 34.8 | 57.86 | 29.9 |

## A.12 LIMITATIONS

Despite the strong empirical performance of KBVQ-MoE on ultra–low-bit quantization of decoder-only MoE large language models, the method exhibits several limitations that highlight potential directions for future work.

(1) Dependence on empirically selected SVD truncation rank. The truncated rank (k) in IDRE is currently chosen based on an empirical balance between reconstruction fidelity and storage overhead (e.g., $k = \frac{1}{128}$ of the full rank in our experiments). Although this choice works robustly across evaluated models, KBVQ-MoE does not yet include an adaptive mechanism to automatically determine the optimal rank for different MoE architectures, task regimes, or input statistics. This limits its ability to fully optimize the trade-off between redundancy removal and quantization efficiency across diverse settings.

(2) Limited validation beyond decoder-only architectures. Our experiments focus primarily on decoder-only MoE models such as Qwen-MoE and Mixtral-8×7B. While this allows for a clean examination of the effects of IDRE and BCOS, the method has not yet been systematically evaluated on encoder–decoder MoE models or multimodal MoE architectures. These settings may require modified input-statistics estimation or revised forms of redundancy modeling. Extending KBVQ-MoE to bidirectional or cross-modal MoE structures is therefore an important area for future exploration.

(3) Lack of evaluation in extreme bit regimes. KBVQ-MoE achieves near–FP16 accuracy at 2–3 bits, but has not been tested in more extreme quantization regimes such as 1-bit binary quantization or hybrid bit-widths (e.g., 1.5-bit). These regimes pose substantially greater information bottlenecks, and may require enhanced bias-correction mechanisms, more expressive codebook structures, or hybrid compression strategies to remain effective.

Together, these limitations delineate the boundaries of the current investigation and point toward promising extensions of KBVQ-MoE in future research.

