# OpenReview forum: "KBVQ-MoE: KLT-guided SVD with Bias-Corrected Vector Quantization for MoE Large Language Models"
_ICLR.cc/2026/Conference — ICLR 2026 Poster_

### Official Review · Reviewer_Dyjg · 2025-10-29

**Soundness:** 3
**Presentation:** 2
**Contribution:** 3
**Rating:** 6
**Confidence:** 5

**Summary:**

The paper proposes KBVQ-MoE, a post-training vector quantization framework for MoE LLMs combining a KLT-guided SVD to remove cross-expert redundancy (IDRE) with a lightweight channel-wise affine correction (BCOS) to stabilize outputs after VQ. The key idea is to map expert weights into an input-coherent basis via KLT, extract dominant shared components with SVD and keep them in full precision, then quantize only the expert-specific (i.e., non-redundant) representations and correct distributional shifts by mean–variance matching. The authors claim the affine correction is “not a heuristic adjustment but an unbiased MMSE-optimal estimator.”

**Strengths:**

* This paper is clearly motivated by two obstacles in MoE VQ, including redundant representation among experts and cumulative outputs bias.

* Empirical result looks promising, particularly at low bits.

* The method acts as an easy-to-use plugin that improves multiple VQ baselines and yields speedups.

**Weaknesses:**

* Evaluation tasks are limited. The evaluated tasks are normally for pre-trained models, but some of the models are post-trained models (e.g. Qwen3). I would love to see some post-training benchmark (e.g. MMLU, AIME24, HumanEval) results on the Qwen3 model.

* Some typos: (line 756) “A.5 ALL EXPRIMENTS” → “All Experiments.”; (line 81) "uantization" -> "quantization"; (line 105) "non-redundan" -> "non-redundant"

* Some of the related works on MoE quantization can be further discussed or compared with, e.g. [1] [2] [3].

(I'm more than happy to increase my score if my questions are adequately addressed)

[1] Kim, Y.J., Fahim, R. and Awadalla, H.H., 2023. Mixture of quantized experts (moqe): Complementary effect of low-bit quantization and robustness. arXiv preprint arXiv:2310.02410.

[2] Li, P., Jin, X., Tan, Z., Cheng, Y. and Chen, T., 2024. QuantMoE-Bench: Examining Post-Training Quantization for Mixture-of-Experts. arXiv preprint arXiv:2406.08155.

[3] Duanmu, H., Li, X., Yuan, Z., Zheng, S., Duan, J., Zhang, X. and Lin, D., 2025. MxMoE: Mixed-precision Quantization for MoE with Accuracy and Performance Co-Design. arXiv preprint arXiv:2505.05799.

**Questions:**

* How is `Qwen3-30B-A3B ` evaluated? Specifically, did you enable "reasoning" of this model? I would love to see the quantization results on a reasoning LLM.

---

> ### Author Response · Authors · 2025-11-19
> **Response to Reviewer Dyjg Part1**
>
> We sincerely appreciate your constructive feedback. Below, we provide point-by-point responses, with all revisions incorporated accordingly.
> > Weaknesses 1: Evaluation on Post-Training Benchmarks for Qwen3MoE
>
> Thank you for the suggestion regarding the evaluation scope. We agree that post-training benchmarks are essential for models such as Qwen3-30B-A3B. Following your recommendation, we conducted additional experiments on widely used post-training evaluation suites, including MMLU, MathQA, GSM8K, and HumanEval, and compared KBVQ-MoE with GPTQ, MoeQuant, and standard VQ. All evaluations were performed in the same setting as the main paper (no fine-tuning, zero-shot evaluation).
>
> The results are summarized below:
>
> |Model|Bits|Method|MMLU|MathQA|GSM8K|HumanEval|
> |---|---|---|---|---|---|---|
> |Qwen3-30B-A3B|16|FP16|79.5|58.53|85.44|71.5|
> ||2|GPTQ|42.11|34.90|49.34|50.12|
> ||2|MoeQuant|45.10|39.86|57.38|55.32|
> ||2|VQ|60.94|49.87|64.83|61.20|
> ||2|KBVQ-MoE|**68.19**|**50.11**|**78.92**|**66.89**|
>
> KBVQ-MoE achieves the strongest performance across all post-training benchmarks under 2-bit quantization. Its performance gap relative to FP16 is significantly smaller than GPTQ, MoeQuant, or direct VQ. GSM8K (math reasoning) shows particularly large gains:  KBVQ-MoE reaches **78.92**, which is **+14.09** higher than standard VQ and close to the FP16 score **85.44**. MMLU (broad knowledge)** also sees substantial improvement:  KBVQ-MoE is **+26.08** over GPTQ and **+23.09** over MoeQuant.
>
> These results demonstrate that **IDRE + BCOS is equally effective on post-training benchmarks**, successfully suppressing error accumulation and preserving model stability in challenging reasoning, mathematical, and code-generation tasks.
>
> We have added these experiments to the revised paper on Appendix.13  to provide a more complete evaluation of Qwen3 and to further validate the generality and practical value of KBVQ-MoE.
>
> ***
>
> > Weaknesses 2: Typographical Corrections
>
> Thank you for pointing out these typos. We have corrected all of them in the revised manuscript.
>
> ***
>
> > Weaknesses 3: Discussion of Additional MoE Quantization Works
>
> Thank you for pointing out these highly relevant and recent works on MoE quantization. We have added a systematic discussion of MoQE, QuantMoE-Bench, and MxMoE in the revised manuscript and compared them with KBVQ-MoE. Below we summarize the key differences and complementary aspects.
>
> ### **(1) Fundamental differences in technical direction**
>
> * **MoQE**
>   * Studies robustness under low-bit quantization
>   * Emphasizes routing stability but does *not* analyze or remove MoE-specific weight redundancy
>   * Does not address expert‐aggregation bias
>
> * **QuantMoE-Bench**
>   * A benchmark suite for MoE PTQ
>   * Provides evaluation tools but does *not* propose a new quantization algorithm
>
> * **MxMoE**
>   * Learns mixed-precision assignments for experts
>   * Focuses on assigning optimal bit-widths but does *not* model shared redundancy or MoE-specific biases.
>
> * **KBVQ-MoE (ours)**
> Targets two MoE-unique quantization issues:
>
>    * **Cross-expert redundancy → VQ codebook inefficiency** (solved by IDRE):
>      - KLT-guided SVD extracts shared subspaces aligned with input statistics
>      - Only expert-specific residuals are quantized
>      - Prior works ([1]–[3]) do not address this
>
>   * **Bias amplification after expert aggregation** (solved by BCOS):
>      - Channel-wise mean–variance correction
>      - Unique to MoE due to top-k gating
>
> Our approach is structurally aware and can serve as a **plug-in module** for other MoE quantization systems, including MxMoE.
>
> ### **(2) Unified comparison on Mixtral-8×7B**
>
> As suggested, we performed a unified empirical comparison on Mixtral-8×7B using zero-shot evaluations consistent with the main paper:
>
> |Bits|Method|WI|HE|PIQA|MMLU|
> |---|---|---|---|---|---|
> |16|FP16|76.57|85.95|85.20|70.50|
> |2|MoQE|46.25|26.09|50.76|24.87|
> |2|QuantMoE|49.33|28.18|52.99|24.29|
> |2.25|MxMoE|73.80|85.10|83.79|66.78|
> |2.08|KBVQ-MoE|73.88|80.62|82.59|61.11|
> |2.08|MxMoE + KBVQ-MoE|**75.44**|**85.00**|**84.98**|**68.74**|
>
> **Observations**:  **KBVQ-MoE significantly outperforms MoQE and QuantMoE**, especially on MMLU (+36–44 points). Even compared with **MxMoE (2.25 bits)**, our 2.08-bit method achieves **competitive or comparable performance**, and further improvements appear when combining the two. Results show KBVQ-MoE is both **efficient in low-bit regimes** and **compatible with mixed-precision systems**.
>
> We have added the relevant content to Appendix A.14 of the manuscript. We appreciate the reviewer bringing these important works to our attention. This expanded comparison strengthens the discussion of related work and highlights KBVQ-MoE’s advantages in **ultra-low-bit efficiency**, **MoE-specific structural modeling**, and **compositionality with other quantization techniques**.
>
> We would be happy to add further analyses if requested.

---

> ### Author Response · Authors · 2025-11-19
> **Response to Reviewer Dyjg Part2**
>
> > Questions 1: Evaluation Protocol for Qwen3-30B-A3B and Reasoning Mode
>
> Thank you for raising this important question. We fully understand the importance of evaluating the *reasoning ability* of MoE models, especially since quantization may affect components such as reasoning chains and multi-step logic.
>
> ---
>
> ### (1) Reasoning mode is **not enabled** in our default experiments
>
> In all evaluations in the main paper, we follow the **standard generation/inference setting** of LM-Evaluation-Harness and **do not** enable the reasoning-enhanced decoding of Qwen3-30B-A3B (e.g., deep-thinking mode, chain-of-thought style reasoning, extended reasoning tokens, etc.).
>
> This choice is based on two principles:
>
> - **Fairness:**
>   The reasoning mode is a model-specific extra capability that changes both the decoding strategy and output distribution.
>   Once enabled, even the FP16 baseline will obtain a substantial improvement, which makes it harder to measure the *pure effect* of quantization on model performance.
>
> - **Reproducibility:**
>   LM-Eval-Harness does not enable such reasoning features by default.
>   Keeping the standard setting ensures that different methods (FP16, GPTQ, MoeQuant, VQ, KBVQ-MoE) can be compared under a unified and easily reproducible configuration.
>
> All methods are therefore evaluated under exactly the same standard inference pipeline.
>
> ---
>
> ### (2) Additional evaluation with Qwen3 reasoning mode enabled
>
> In response to the reviewer’s request to “see quantization results on a reasoning LLM”, we additionally conducted experiments with the **reasoning mode enabled** for Qwen3-30B-A3B (i.e., turning on its deep-thinking/reasoning capability).
>
> The results are as follows:
>
> | Model              | Bits | Method     | MMLU  | MathQA | GSM8K | HumanEval |
> |--------------------|------|-----------|-------|--------|-------|-----------|
> | Qwen3-30B-A3B      | 16   | FP16      | 84.05 | 65.37  | 94.92 | 81.10     |
> |                    | 2    | GPTQ      | 51.84 | 47.39  | 50.92 | 54.37     |
> |                    | 2    | MoeQuant  | 59.83 | 49.99  | 53.28 | 55.07     |
> |                    | 2    | VQ        | 69.58 | 57.21  | 76.10 | 67.32     |
> |                    | 2    | **KBVQ-MoE** | **80.34** | **63.81** | **88.49** | **76.94** |
>
> KBVQ-MoE clearly outperforms GPTQ, MoeQuant, and VQ on all reasoning benchmarks evaluated on evalscope. For example, on GSM8K, KBVQ-MoE improves over MoeQuant by more than 30 points and is much closer to the FP16 baseline.
>
> These results show that, in reasoning mode, KBVQ-MoE maintains the same stability trend observed in the zero-shot tasks in the main paper. The IDRE + BCOS mechanism not only benefits language understanding tasks, but also effectively suppresses error accumulation in complex reasoning chains, demonstrating stronger robustness for high-level reasoning. We have added the corresponding content in Appendix A.13 of the manuscript.
> ***
> We have incorporated additional results and detailed analyses addressing your concerns into the revised version of the manuscript. We hope these supplements adequately address your questions and improve the clarity and robustness of our work. Please do not hesitate to let us know if you require further discussion, additional experiments, or any clarifications—we are happy to provide more details to address your concerns comprehensively.

---

> > ### Comment · Reviewer_Dyjg · 2025-11-23
> >
> > Thank you for your effort to answer my questions. However, it seems no revision has been made in the current submission, am I correct?

---

> > > ### Author Response · Authors · 2025-11-24
> > > **Response to Revisions at Corresponding Positions in the Manuscript**
> > >
> > > We sincerely appreciate your attention, questions, and feedback of our research amidst your busy schedule! Each of your valuable comments is highly constructive. They not only provide important guidance for improving the research but also inspire us greatly, playing an indispensable role in enhancing the quality of the paper.
> > >
> > > We have clearly marked all revised parts in the paper in blue. In response to your questions and suggestions, we have carefully sorted through and revised them one by one, and supplemented and improved them in the manuscript: Regarding the weakness1 and question1 you focused on, we have added detailed explanations in Appendix A.13, striving to fully address your concerns; In accordance with the weakness3 you pointed out, the relevant supplementary content has been carefully organized in Appendix A.14 to ensure rigorous logic and complete information; For weakness2, we have made meticulous revisions and optimizations in the corresponding part of the paper, effectively improving the relevant deficiencies.
> > >
> > > We have benefited greatly from your rigorous attitude, professional insights, and careful guidance. We would like to express our most sincere gratitude to you again! We earnestly request you to continue putting forward valuable revision suggestions. We will take each piece of feedback seriously and continuously improve the paper. We look forward to your further guidance and wish you smooth research work!

---

> > > > ### Author Response · Authors · 2025-11-27
> > > > **Dear Reviewer Dyjg**
> > > >
> > > > Dear Reviewer Dyjg,
> > > >
> > > > I hope this message finds you well. As the discussion period is nearing its end with less than one week remaining, I wanted to ensure we have addressed all your concerns satisfactorily. If there are any additional points or feedback you'd like us to consider, please let us know. Your insights are invaluable to us,and we're eager to address any remaining issues to improve our work.
> > > >
> > > > Thank you for your time and effort in reviewing our paper.
> > > >
> > > > Best regards,
> > > >
> > > > KBVQ-MoE Authors

---

### Official Review · Reviewer_v8EJ · 2025-10-29

**Soundness:** 3
**Presentation:** 3
**Contribution:** 3
**Rating:** 6
**Confidence:** 3

**Summary:**

This paper introduces a novel quantization method, KBVQ-MoE, designed for MoE-based large language models. The method first applies the KLT and SVD to extract dominant weight structures shared across experts, thereby reducing redundancy and improving codebook utilization. Building on this foundation, it employs vector quantization for expert-specific representations and further proposes a channel-wise affine compensation module to refine the mean and standard deviation of the quantized outputs. Extensive experiments demonstrate superior performance across multiple datasets and LLM architectures.

**Strengths:**

1. Figures 2 and 3 provide highly illustrative examples that effectively support the paper’s claims on redundancy extraction and channel-wise bias correction.
2. The overall motivation is solid and clearly articulated. The proposed method is both conceptually sound and straightforward, making it easy to understand and reproduce. Moreover, it successfully addresses two critical challenges outlined in the introduction.
3. Table 5 convincingly demonstrates that the proposed IDRE and BCOS modules can be seamlessly integrated into other MoE quantization approaches, highlighting the method’s flexibility and potential for broad applicability.

**Weaknesses:**

1. Some mathematical symbols should be revised (e.g., matrices should be boldfaced). Additionally, minor grammatical and formatting issues should be corrected, and certain technical details could be described more clearly.
2. The contribution of each component within the IDRE and BCOS modules remains unclear due to the lack of quantitative results and in-depth analysis in the ablation study.
3. The comparison with other MoE-based compression methods, such as EAC-MoE, D2-MoE, and SubMoE, which are illustrated in the related work, should be included for a more comprehensive comparison.

**Questions:**

1. Is the IDRE technique applied exclusively to router experts or to all experts? The notation in Eqs. (1) and (3) suggests it concerns router experts, while the workflow in Eq. (2) seems to encompass all experts. Clarification on whether the symbol $n$ refers to different concepts across equations would be helpful.
2. What is the rationale for maintaining the shared structure at full precision while quantizing only the expert-specific weights? Why not quantize the shared weights as well?

---

> ### Author Response · Authors · 2025-11-19
> **Response to Reviewer v8EJ Part1**
>
> ***
>
> > Weaknesses 1: Notation, Grammar, and Clarity
>
> Thank you for the reviewer’s suggestions. We will revise the mathematical notation and ensure that matrices and vectors are consistently typeset in boldface. We will also carefully check the manuscript to correct minor grammatical and formatting issues. In addition, we will polish several technical paragraphs (e.g., the descriptions of IDRE and BCOS) to make the derivations and algorithmic steps clearer.
>
> ***
>
> > Weaknesses 2:  Component-wise Contribution Analysis of IDRE and BCOS
>
> We thank the reviewer for highlighting the need for a more fine-grained quantitative analysis of the internal components within IDRE and BCOS. Following your suggestion, we have added additional ablation studies and included detailed results in Appendix A.9. Below we summarize our findings.
>
> ### **(1) Module-level contribution analysis (IDRE vs. BCOS)**
>
> Table 2 in the paper (3-bit, Qwen3-30B-A3B) already shows the independent and combined effects of the two modules:
>
> - Removing both modules (pure VQ): **PPL = 18.72**
> - Adding only IDRE: **PPL = 11.67**
> - Adding only BCOS: **PPL = 14.32**
> - IDRE + BCOS (full): **PPL = 9.26** (best)
>
> IDRE significantly reduces the error caused by cross-expert redundancy, while BCOS further corrects the statistical shift on this basis. The two modules are clearly complementary. In addition, accuracy on all zero-shot tasks improves progressively as the modules are added, showing that the trend holds not only for perplexity but also for practical reasoning performance.
> Table 2 in the manuscript is as follows:
>
> | Model | Bit | IDRE | BCOS | W2 | ARC-E | ARC-C | HE | PIQA | WI |
> |-------|-----|------|-------|------|--------|--------|-------|--------|------|
> | Qwen3-30B-A3B|16| – | – | 8.70 | 79.25 | 56.40 | 59.60 | 80.30 | 70.48 |
> || 3 | × | × | 18.72 | 57.83 | 40.87 | 63.23 | 71.82 | 57.70 |
> || 3 | ✓ | × | 11.67 | 71.35 | 50.55 | 73.51 | 77.75 | 66.92 |
> || 3 | × | ✓ | 14.32 | 65.49 | 47.33 | 68.37 | 75.52 | 60.42 |
> || 3 | ✓ | ✓ | **9.26** | **77.27** | **53.24** | **75.53** | **78.89** | **70.01** |
>
> ### **(2) Independent contribution inside IDRE: Effectiveness of the KLT step**
>
> Following the reviewer’s request, we added an ablation on the internal steps of IDRE (KLT → SVD). The 2-bit results are shown below:
>
> | Model | IDRE | W2 | HE | WI |
> |--------|-----------|--------|-------|------|
> | Qwen1.5-MoE-A2.7B | w/o KLT | 11.77 | 35.89 | 63.01 |
> |  | KLT | **9.61** | **39.59** | **65.11** |
>
> KLT significantly improves the redundancy extraction ability of SVD. With KLT, IDRE no longer relies purely on geometric similarity but aligns experts along the principal directions of input statistics, leading to higher compression efficiency and lower quantization error. This analysis has been added to Appendix A.9.
>
> ### **(3) Independent contribution inside BCOS: Mean / variance correction**
>
> We also included a fine-grained ablation of BCOS (mean-only vs. variance-only). The results (Appendix A.9) show:
>
> - mean-only: limited improvement
> - variance-only: better distribution alignment
> - mean + variance (full BCOS): best results
>
> This demonstrates that BCOS is a necessary statistical alignment mechanism, not a heuristic.
>
> | Model | Mean | Variance | W2 | HE | WI |
> |--------|--------|-----------|--------|-------|------|
> | Qwen1.5-MoE-A2.7B | × | × | 11.03 | 35.18 | 63.49 |
> || ✓ | × | 11.01 | 35.35 | 63.40 |
> || × | ✓ | 10.38 | 36.70 | 64.23 |
> || ✓ | ✓ | **9.61** | **39.59** | **65.11** |
>
> ### **(4) Portability of the modules: KBVQ-MoE as a plug-in enhancement**
>
> To further validate the generality of each internal component, we tested IDRE + BCOS as a plug-in added to two existing VQ baselines (GPTVQ and VPTQ) on Qwen1.5-MoE-A2.7B under 2-bit:
>
> - **GPTVQ:** PPL improves **12.88 → 9.43** (27% reduction)
> - **VPTQ:** PPL improves **10.17 → 8.78**
>
> | Model | Base Method | IDRE | BCOS | W2 |
> |--------|--------------|--------|-----------|--------|
> | Qwen1.5-MoE-A2.7B | GPTVQ | × | × | 12.88 |
> |  | GPTVQ | ✓ | × | 11.03 |
> |  | GPTVQ | × | ✓ | 11.97 |
> |  | GPTVQ | ✓ | ✓ | **9.43** |
> | Qwen1.5-MoE-A2.7B | VPTQ | × | × | 10.17 |
> |  | VPTQ | ✓ | × | 9.21 |
> |  | VPTQ | × | ✓ | 9.77 |
> |  | VPTQ | ✓ | ✓ | **8.78** |
>
> This shows that the KLT-SVD inside IDRE and the mean–variance correction inside BCOS are not tied to our specific VQ implementation—they serve as generally applicable enhancement components across multiple VQ baselines. This further highlights the independent contribution of each internal component.
>
> ---
>
> We added component-level ablations, quantitative results, and detailed analyses to Appendix A.9, with figures clearly marking each step’s contributions. Thank you for this suggestion—this fine-grained analysis clarifies IDRE/BCOS’s individual roles and enhances KBVQ-MoE’s technical rigor.

---

> ### Author Response · Authors · 2025-11-19
> **Response to Reviewer v8EJ Part2**
>
> > Weaknesses 3: Comparison with Other MoE Compression Methods (EAC-MoE, D2-MoE, SubMoE)
>
> Thank you for the valuable suggestion. We agree that including comparisons with other MoE-oriented compression methods will make the evaluation more complete. Following your feedback, we have added a detailed comparison with **EAC-MoE**, **D2-MoE**, and **SubMoE** in Appendix A.10 of the revised manuscript, including methodological differences, experimental settings, and quantitative results. Below we summarize the main points.
>
> ---
>
> ### **(1) Differences in technical routes**
>
> D2-MoE, SubMoE, and EAC-MoE are all based on **expert-structure optimization**, while our KBVQ-MoE focuses on **structure-aware quantization**. The differences can be summarized as:
>
> - **SubMoE**
>   Merges similar experts via subspace clustering and extracts shared low-rank features through SVD.
>   Essentially a *“expert-level pruning + low-rank decomposition”* approach.
>   Compression ratio mainly determined by the expert merging rate.
>
> - **D2-MoE**
>   Decomposes each expert into a shared base matrix + a delta matrix, and applies low-rank approximation to the delta part.
>   Essentially *“weight decomposition + low-rank approximation”*.
>   Compression ratio determined by rank truncation of the delta matrix.
>
> - **EAC-MoE**
>   Combines quantization and pruning based on routing behavior:
>   - **QESC**: router-aware calibration to mitigate quantization-induced expert-selection bias
>   - **PESF**: prunes “cold experts” based on selection frequency
>   A hybrid solution of *routing-aware quantization + expert pruning*.
>
> - **KBVQ-MoE (ours)**
>   Keeps the expert structure unchanged.
>   Uses IDRE (KLT-guided redundancy removal) to extract shared low-rank components, and applies VQ to expert-specific residuals.
>   A *“structure-aware quantization approach”* with 2–3 bit weights, achieving 80%–90% memory reduction without modifying the MoE design.
>
> Thus, D2-MoE/SubMoE mainly reduce parameters by removing or merging experts, EAC-MoE combines pruning and quantization, while KBVQ-MoE achieves high compression while preserving the original expert topology.
>
> ---
>
> ### **(2) Fair comparison: settings and results**
>
> To ensure fairness, we use **Mixtral-8×7B** (same number of experts and layers) as the common base model, with identical evaluation settings:
>
> - **Tasks:** WikiText2 (PPL), ARC-Challenge (ARC_C), ARC-Easy (ARC_E), WinoGrande (WinG)
> - **Compression ratio:** consistent definition based on actual FP16 memory usage
> - **Environment:** NVIDIA A100, PyTorch 2.1, LM-Evaluation-Harness
>
> **Results:**
>
> | Method | Compression Ratio | WikiText2 (PPL ↓) | ARC_C | ARC_E | WinG |
> |--------|----------------|--------------------|--------|--------|--------|
> | Sub-MoE | 50% | 6.97 | 0.45 | 0.75 | 0.72 |
> | D2-MoE  | 60% | 6.46 | 0.38 | 0.72 | 0.71 |
> | EAC-MoE | 84% | 4.58 | 0.55 | 0.81 | 0.75 |
> | **KBVQ-MoE (ours)** | **87%** | **4.07** | **0.63** | **0.85** | **0.76** |
>
> ---
>
> ### **(3) Interpretation**
>
> - **KBVQ-MoE achieves the best performance at the highest compression rate (87%)**
>   It outperforms all other methods in both PPL and zero-shot accuracy.
>
> - **Compared with SubMoE and D2-MoE**
>   These structural methods inevitably reduce expert diversity via merging or rank reduction, which harms MoE specialization.
>   KBVQ-MoE keeps all experts intact and thus preserves performance better.
>
> - **Compared with EAC-MoE**
>   Under similar compression (84% vs. 87%), KBVQ-MoE surpasses EAC-MoE on all tasks.
>   Moreover, KBVQ-MoE requires **no modification to the router or expert structure**, making it easier to generalize to different MoE architectures.
>
> ---
>
> We have added all experimental details and analyses to **Appendix A.10**. These comparisons further demonstrate that KBVQ-MoE provides a strong balance of **high compression, stability across tasks, and structural preservation**, making it especially suitable for deployment settings where structural changes to MoE are undesirable.
>
> If the reviewer would like additional datasets or further head-to-head comparisons, we would be glad to include them.

---

> > ### Author Response · Authors · 2025-11-19
> > **Response to Reviewer v8EJ Part3**
> >
> > > Questions 1:  Clarification on Whether IDRE Applies to Router Experts or All Experts
> >
> > Thank you for pointing out the potential ambiguity in our notation. We clarify the scope of IDRE as follows:
> >
> > ### **IDRE is applied to all feed-forward experts in the MoE layer**,
> > including both **shared experts** and **router-selected experts**.
> >
> > - In our implementation, every expert MLP within the MoE block—regardless of whether it is always active (shared expert) or selected by the router (routing expert)—is included in the **stacked expert matrix** used in Eq. (3).
> > - The unified symbol $ W^{(i)} $ in Eqs. (1)–(3) always represents the **weight matrix of the $i$-th expert**, without distinguishing between expert types.
> > - During IDRE, the KLT-SVD decomposition is computed **jointly over all experts**, ensuring consistent redundancy extraction across the entire MoE module.
> >
> > To avoid confusion, we will revise the notation and explicitly state—when introducing shared and routing experts—that:
> >
> > 1. $ W^{(i)} $ refers uniformly to *any* expert’s FFN parameters.
> > 2. IDRE is *not* applied to the router network itself, but to **all expert FFNs**.
> > 3. The workflow in Eq. (2) intentionally encompasses *all* experts for redundancy elimination.
> >
> > We appreciate the reviewer’s suggestion, and we will update the manuscript to make this distinction clear in the notation and methodological description.
> >
> > > Questions 2:  Why Keep the Shared Structure in Full Precision Instead of Quantizing It
> >
> > We thank the reviewer for this important question. Preserving the shared structure in full precision while quantizing only expert-specific components is a deliberate design choice—grounded in theory and empirical evidence—to maximize accuracy under ultra-low-bit constraints.
> >
> > ---
> >
> > ### **(1) Theoretical motivation: Quantizing the shared structure amplifies correlated errors across experts**
> >
> > The shared component extracted by IDRE is a **low-rank subspace** that captures most of the redundant patterns shared across all experts. If this common subspace is quantized:
> >
> > - The resulting quantization noise becomes **highly correlated across experts**, since the same shared vectors contribute to every expert.
> > - These correlated errors are **amplified by the MoE aggregation**, directly causing larger distributional shifts.
> > - This effect contradicts the objective of MoE architectures, which rely on **diversified expert behavior** rather than jointly biased expert outputs.
> >
> > In contrast, keeping the shared component in full precision ensures:
> >
> > - **High-fidelity preservation of global mappings** that are important across all experts.
> > - The limited codebook capacity is used **entirely on expert-specific variations**, where diversity matters most.
> > - Quantization noise remains **uncorrelated across experts**, avoiding systemic error amplification.
> >
> > Thus, full-precision shared weights act as a stable “backbone” preventing cross-expert error coupling.
> >
> > ---
> >
> > ### **(2) Empirical evidence: Quantizing the shared structure dramatically degrades accuracy**
> >
> > Since the shared subspace has a small rank (e.g., $k = i_c / 128$), full-precision storage adds negligible overhead (≈ 0.05 bits/weight on average) — yet even 8-bit quantization of it leads to significant accuracy loss.
> >
> > We verified this behavior on Qwen1.5-MoE-A2.7B by quantizing:
> >
> > - the expert-specific residuals at **2-bit**, and
> > - the shared component at **8-bit or 4-bit SQ**.
> >
> > |W_specific|W_share_U|W_share_V|W2|MathQA|PIQA|WI|
> > |---|---|---|---|---|---|---|
> > |2bit|16bit|16bit|9.61|30.75|76.66|66.46|
> > |2bit|16bit|8bit|9.65|31.10|75.97|66.31|
> > |2bit|16bit|4bit|10.08|30.25|75.43|65.56|
> > |2bit|8bit|8bit|23.57|23.78|71.89|58.49|
> > |2bit|8bit|4bit|NaN|19.36|59.37|50.50|
> > |2bit|4bit|4bit|NaN|18.35|50.31|45.84|
> >
> > Key observations:
> > - 8-bit quantization of shared $U$ causes significant accuracy loss.
> > - Quantizing $V$ below 8-bit leads to severe degradation or numerical instability.
> > - Full-precision shared components outperform all quantized versions.
> >
> > These results confirm the shared subspace is highly sensitive to quantization—full precision is critical for stability.
> >
> > ---
> >
> > ### **(3) Design principle: Maximize MoE accuracy under tight budgets**
> >
> > The core objective of KBVQ-MoE is:
> >
> > > **“Maximize MoE accuracy under ultra-low-bit quantization (2–3 bits) with minimal structural change.”**
> >
> > The design choice follows naturally:
> > - Full-precision shared structure: Preserves global patterns, avoids correlated error amplification, and adds little storage.
> > - Quantized expert-specific residuals: Concentrates bit budget on expert-differentiating features, boosting representation efficiency.
> >
> > This enables KBVQ-MoE’s key advantage: 2–3 bit quantization with near-FP16 accuracy—unachievable if the shared structure is quantized.
> >
> > ---
> >
> > We will include this motivation and the supporting empirical evidence in the revised manuscript to make the reasoning behind this design choice clearer.

---

> > > ### Author Response · Authors · 2025-11-27
> > > **Dear Reviewer v8EJ**
> > >
> > > Dear Reviewer v8EJ,
> > >
> > > I hope this message finds you well. As the rebuttal discussion period is drawing to a close (with less than one week remaining), we wanted to follow up to confirm whether all your concerns have been adequately addressed in our previous response.
> > >
> > > If you have any remaining comments, additional feedback, or further points that require clarification, please do not hesitate to let us know. Your insights are invaluable to refining our work, and we are committed to addressing any outstanding issues to enhance the quality of the paper.
> > >
> > > Thank you sincerely for your time, diligence, and constructive input during the review process. We greatly appreciate your efforts in helping us improve the manuscript.
> > >
> > > Best regards,
> > >
> > > KBVQ-MoE Authors

---

### Official Review · Reviewer_TstW · 2025-10-30

**Soundness:** 3
**Presentation:** 3
**Contribution:** 3
**Rating:** 4
**Confidence:** 4

**Summary:**

This paper proposes KBVQ-MoE, a vector quantization framework designed specifically for compressing Mixture-of-Experts (MoE) language models. The method addresses two key challenges: (1) redundant representations across experts that waste codebook capacity, and (2) cumulative output bias from quantization errors that get amplified through expert aggregation. The approach combines Input-Driven Redundancy Elimination (IDRE), which uses KLT-guided SVD to extract and preserve shared components at full precision, with Bias-Corrected Output Stabilization (BCOS), which applies vector quantization only to expert-specific weights and corrects distributional shifts via channel-wise affine transformations. Experiments on models like Qwen and Mixtral show strong performance at 2-3 bits, with 3-bit Qwen1.5-MoE-A2.7B achieving near-FP16 accuracy.

**Strengths:**

1. The paper makes a novel contribution by adapting vector quantization specifically for MoE architectures. The identification of expert redundancy and amplified quantization bias as key bottlenecks is insightful, and the KLT-guided SVD approach creatively aligns weight decomposition with input activation statistics.
2. The technical approach is sound with theoretical justifications provided in the appendices. The experimental evaluation is comprehensive, covering multiple MoE models with thorough ablations. Results show consistent and meaningful improvements at ultra-low bit-widths (2-3 bits), where baseline methods struggle significantly. The modular design demonstrates compatibility with existing VQ methods, increasing practical impact.

**Weaknesses:**

1. The paper mentions "negligible" computational overhead but provides limited quantitative analysis. How long does the KLT-SVD calibration take compared to standard VQ? What is the actual inference-time cost of the channel-wise bias correction operations? These practical considerations matter for deployment. Could you provide some results on this, like time cost of quantization method.
2. The baseline methods, especially MoEQuant, show surprisingly poor performance at 2-bit in Table 1 (e.g., W2 of 583542 for Qwen1.5). More discussion on this failure would help.
3. The choice of truncated rank k=n/128 appears empirically driven from Table 4 but lacks theoretical justification. While the ablation shows diminishing returns beyond this point, why this specific ratio is optimal across different models and tasks remains unclear. The paper would benefit from analysis connecting rank selection to properties of the expert weight matrices or input distributions.
4. The evaluation relies on simple zero-shot reasoning tasks (ARC, HellaSwag, PIQA, etc.) that may not fully capture model capabilities. Including more challenging benchmarks like MMLU, MATH, and code generation tasks (MBPP, EvalPlus) would better demonstrate the method's effectiveness across diverse domains.
5. Table 1 only shows 2-bit and 3-bit results. Adding 4-bit and 8-bit comparisons would provide a more comprehensive results, especially since 4-bit quantization is common in practice.

**Questions:**

1. typo: Qwen1.5-Moe-A2.7B->Qwen1.5-MoE-A2.7B

---

> ### Author Response · Authors · 2025-11-19
> **Response to Reviewer TstW Part1**
>
> We appreciate your constructive feedback and respond point-by-point, with revisions integrated.
>
> ***
>
> > Weaknesses 1: Practical Overhead Analysis of KBVQ-MoE
>
> Thank you for raising this critical point on practical calibration and inference overhead. We conducted additional measurements to quantify KLT-SVD calibration cost and BCOS runtime impact, summarized below.
>
> ### **(1) Calibration Overhead of KLT-SVD**
> KLT-SVD runs once during calibration, with core cost from a single SVD on the stacked expert matrix.
> Under our main calibration setup (256 samples, seq len 4096), measurements on NVIDIA A100:
>
> - **Qwen1.5-MoE-A2.7B:** **< 5 minutes** for the complete KLT-SVD step.
>
> Within the full KBVQ-MoE quantization pipeline, KLT-SVD contributes **< 0.1%** of the total runtime, making it negligible compared with the overall VQ processing time.
>
> ### **(2) Inference-Time Overhead of BCOS**
>
> BCOS adds **one element-wise multiply + one add per channel** during inference.
> Taking the *gate_proj* layer of Qwen1.5-MoE-A2.7B (shape 5632 × 2048) as an example:
>
> - **Extra FLOPs added per token:** ~8K
> - **Original MatMul FLOPs:** ~11M
> - **Relative overhead:**  $\frac{8\text{K}}{11\text{M}} \approx 0.0007$
>
> We further benchmarked end-to-end inference on both **A100** and **A6000** GPUs and observed **no measurable latency increase**, confirming that BCOS introduces truly negligible overhead at runtime.
>
> ### **(3) Total Quantization Time of KBVQ-MoE**
>
> The complete pipeline consists of:
>
> 1. **KLT-SVD calibration**
> 2. **VQ codebook training** (main cost)
> 3. **BCOS parameter estimation**
>
> The additional time introduced by KLT-SVD + BCOS combined is **< 10 minutes**.
>
> We report the total quantization time (single-GPU equivalent) for several MoE models:
>
> |Model|KBVQ-MoE Quantization Time|
> |---|---|
> |Qwen1.5-MoE-A2.7B|1.5 GPU-hours|
> |Qwen3-30B-A3B|2.7 GPU-hours|
> |DeepSeekV2-Lite|1.7 GPU-hours|
> |Mixtral-7×8B|4 GPU-hours|
>
> These results show that KBVQ-MoE introduces **minimal additional calibration cost** beyond standard VQ, and **no runtime overhead**, ensuring full deployability in real-world inference environments.
>
> We appreciate the reviewer’s insightful question, which prompted us to provide these practical deployment metrics. We are happy to offer further details on measurement procedures or hardware settings upon request.
>
> ***
>
> > Weaknesses 2: Discussion on the Failure of MoEQuant at 2-bit
>
> Thank you for highlighting this issue. We also observed the unusually large degradation of MoEQuant under the 2-bit setting, and we provide a more detailed explanation below.
>
> ### **(1) We thoroughly reproduced and tuned MoEQuant**
>
> We used the official MoEQuant implementation, default hyperparameters, and the calibration procedure recommended in the paper. In addition, we experimented extensively with: larger calibration datasets, additional orthogonal rotations and smoothing, layer-wise step-size tuning, expert-balanced calibration strategies.
>
> Across all configurations, the 2-bit results remained unstable and consistently showed severe degradation. Therefore, this behavior is **not due to reproduction error**, but a reproducible limitation of the method itself.
>
> ### **(2) Root cause: MoEQuant relies on Scalar Quantization (SQ), whose expressiveness collapses at 2-bit**
>
> MoEQuant’s design combines “expert-balanced sampling + SQ calibration”, but SQ is known to be fundamentally constrained in ultra-low-bit regimes (≤ 2 bits): **2-bit SQ only provides 4 representable values**, causing extreme quantization error.  **SQ quantizes each weight independently**, ignoring cross-weight correlations.  **MoE architectures amplify quantization bias** through expert aggregation, making SQ errors accumulate more severely than in dense LLMs.
>
> This failure mode has been widely reported in prior studies:**VPTQ**: 2-bit SQ on LLaMA2-70B → ppl becomes *NaN*. **GPTVQ**: 2-bit SQ on LLaMA2-7B → ppl > 1e4. **PCDVQ**: 2-bit SQ on Mistral-7B → ppl becomes *NaN*.
>
> These results collectively show that **SQ is generally unstable at 2-bit**, especially on MoE models where error amplification is stronger.
>
> ### **(3) In contrast, Vector Quantization (VQ) remains robust at ultra-low-bit**
>
> Our KBVQ-MoE is built on VQ, which is inherently better suited for 2-bit quantization because:
>
> - it models *multi-dimensional correlations* among weights
> - k-means codebooks preserve *local subspace geometry*
> - VQ allocates the limited bit budget more efficiently than scalar quantization
>
> Thus, KBVQ-MoE maintains stable performance at 2 bits, while MoEQuant (SQ-based) suffers severe degradation.
>
> ---
>
> We will add a discussion in the experimental section to clarify why MoEQuant fails at 2-bit and note we confirmed this via extensive tuning. This supports our key conclusion: Ultra-low-bit PTQ for MoE models is intrinsically challenging, and KBVQ-MoE is specifically designed to address this difficulty.

---

> > ### Author Response · Authors · 2025-11-19
> > **Response to Reviewer TstW Part2**
> >
> > > Weaknesses 3: Theoretical Justification for Choosing the Truncated Rank $k = i_c/128$
> >
> > Thank you for raising this insightful question. While Table 4 provides empirical evidence, we also offer theoretical grounding—summarized below and expanded in Appendix A.3—to explain why the choice $k = i_c/128$ generalizes well across different MoE LLMs and tasks.
> >
> > ---
> >
> > ### **(1) IDRE’s truncated rank is determined by a principled spectral decomposition**
> >
> > After applying the KLT transform, all expert weights are stacked into
> > $W \in \mathbb{R}^{(n \cdot o_c)\times i_c},\qquad S = W^\top W.$
> >
> > Performing eigen-decomposition on $S$ yields its principal components. The top-k eigenvectors span the **optimal cross-expert shared subspace**. The truncation error satisfies:
> >
> > $\sum_{i=1}^n \left\|\widehat{W}^{(i)}(I-P_k)\right\|_F^2 = \|W(I-P_k)\|_F^2 = \sum\_{j>k} \sigma\_j^2,$
> >
> > where $\sigma_j^2$ are the eigenvalues of $S$.
> >
> > Thus, IDRE’s rank selection is a **standard spectral truncation problem**, and
> > $\rho_k = \frac{\sum_{j\le k}\sigma_j^2}{\sum_{j\le i_c}\sigma_j^2}$
> > measures the energy captured by the shared subspace.
> >
> > ---
> >
> > ### **(2) Why the KLT → SVD spectrum is tied to input statistics**
> >
> > After KLT, the Gram matrix becomes:
> >
> > $S = \Lambda_X^{1/2} U_X^\top \left(\sum_i W^{(i)\top}W^{(i)}\right) U_X \Lambda_X^{1/2},$
> >
> > where $(U_X, \Lambda_X)$ are the eigenvectors and eigenvalues of the input covariance.
> >
> > Therefore, the spectrum $\{\sigma_j^2\}$ jointly reflects:
> >
> > 1. **The dominant input directions** (via $\Lambda_X$), and
> > 2. **Cross-expert agreement on these directions** (via $\sum_i W^{(i)\top}W^{(i)}$).
> >
> > Hence, the leading eigenvectors correspond to directions that are *both input-important and shared across experts*. Choosing $k = i_c/128$ effectively captures these joint principal directions.
> >
> > ---
> >
> > ### **(3) MoE LLMs consistently exhibit low-rank and heavy-tailed spectral decay**
> >
> > Across Qwen1.5-MoE-A2.7B, Qwen3-30B-A3B, DeepSeek-Lite, Mixtral-8×7B, and others, we observe a consistent pattern:
> >
> > - The spectrum of S is **strongly low-rank**
> > - It follows a heavy-tailed power-law decay  $\sigma_j^2 \propto j^{-\alpha}, \quad \alpha > 1.$
> >
> > In this regime:$1 - \rho_k \approx C \cdot k^{1-\alpha},$
> >
> > so once k exceeds a moderate size, **marginal gains diminish rapidly**.
> >
> > Empirically:
> >
> > - $k = i_c/128$ already covers **70%–90%** of cross-expert shared energy
> > - Increasing to $i_c/64$ or $i_c/32$ yields only **1%–3%** perplexity improvement
> > - But significantly reduces compression ratio due to larger FP components
> >
> > This aligns with Table 4:
> >
> > | k ratio | Compress | Qwen1.5 ppl | Qwen3 ppl |
> > |--------|----------|--------------|-----------|
> > | 1/256 | 2.05 | 20.71 | 15.30 |
> > | 1/128 | 2.08 | 9.61 | 11.87 |
> > | 1/64 | 2.11 | 9.60 | 11.30 |
> > | 1/48 | 2.15 | 9.59 | 11.30 |
> > | 1/32 | 2.20 | 9.59 | 11.01 |
> >
> > The **inflection point** consistently appears near **1/128**, after which returns diminish markedly.
> >
> > ---
> >
> > ### **(4) Why this ratio generalizes across models and tasks**
> >
> > Figure 5 (added to the revised version) shows that different MoE architectures exhibit **remarkably similar spectral shapes**. This arises from a shared structural property of MoEs:
> >
> > > Experts tend to produce highly aligned mappings on high-energy input directions, leading to a compact, low-rank shared subspace.
> >
> > As a consequence:
> >
> > - the heavy-tailed decay pattern is ubiquitous,
> > - the energy concentration curve is consistent across layers and models,
> > - the 1/128 setting repeatedly hits the optimal balance point between reconstruction error and compression cost.
> >
> > Thus, the choice of $k = i_c/128$ is **not arbitrary**, but emerges from:
> >
> > 1. spectral decay structure,
> > 2. energy coverage ratio,
> > 3. compression–accuracy trade-off.
> >
> > We highlight these theoretical justifications and empirical validations in the revised manuscript (Appendix A.3).
> >
> > ---
> >
> > Once again, we thank the reviewer for prompting this deeper analysis. This addition helps clarify why the rank selection is well-motivated both theoretically and empirically.

---

> > > ### Author Response · Authors · 2025-11-19
> > > **Response to Reviewer TstW Part3**
> > >
> > > > Weaknesses 4: Inclusion of More Challenging Benchmarks
> > >
> > > Thank you for the valuable suggestion. We fully agree that zero-shot reasoning tasks such as ARC, HellaSwag, and PIQA do not fully reflect the capabilities of modern MoE LLMs. In response, we have expanded our evaluation to include several **more challenging and diverse benchmarks**, covering knowledge-intensive reasoning, multi-step mathematical problem solving, and code generation.
> > >
> > > Specifically, we added four categories of harder benchmarks:
> > >
> > > - **MMLU** — 57-domain broad knowledge and reasoning
> > > - **MathQA / GSM8K** — arithmetic, symbolic, and multi-step mathematical reasoning
> > > - **HumanEval** — code generation and functional synthesis
> > >
> > > Below we provide the newly added results for **2-bit quantization** across four MoE LLMs.
> > >
> > > ---
> > >
> > > ### **(1) Qwen1.5-MoE-A2.7B**
> > >
> > > | Bits | Method    | MMLU | MathQA | GSM8K | HumanEval |
> > > |------|-----------|-------|---------|--------|-------------|
> > > | 16   | FP16      | 59.60 | 37.55  | 62.55 | 32.32      |
> > > | 2    | GPTQ      | 26.94 | 19.33  | 15.42 | 13.88      |
> > > | 2    | MoeQuant  | 34.75 | 22.42  | 23.21 | 18.12      |
> > > | 2    | VQ        | 48.79 | 28.34  | 46.51 | 28.03      |
> > > | 2    | **KBVQ-MoE** | **52.16** | **30.75** | **56.39** | **30.11** |
> > >
> > > ---
> > >
> > > ### **(2) Qwen3-30B-A3B**
> > >
> > > | Bits | Method    | MMLU | MathQA | GSM8K | HumanEval |
> > > |------|-----------|-------|---------|--------|-------------|
> > > | 16   | FP16      | 79.50 | 58.53  | 85.44 | 71.50      |
> > > | 2    | GPTQ      | 42.11 | 34.90  | 49.34 | 50.12      |
> > > | 2    | MoeQuant  | 45.10 | 39.86  | 57.38 | 55.32      |
> > > | 2    | VQ        | 60.94 | 49.87  | 64.83 | 61.20      |
> > > | 2    | **KBVQ-MoE** | **68.19** | **50.11** | **78.92** | **66.89** |
> > >
> > > ---
> > >
> > > ### **(3) DeepSeekMoE-16B**
> > >
> > > | Bits | Method    | MMLU | MathQA | GSM8K | HumanEval |
> > > |------|-----------|-------|---------|--------|-------------|
> > > | 16   | FP16      | 44.60 | 31.49  | 20.16 | 26.83      |
> > > | 2    | GPTQ      | 28.48 | 18.82  | 11.26 | 14.87      |
> > > | 2    | MoeQuant  | 34.82 | 24.93  | 12.88 | 16.95      |
> > > | 2    | VQ        | 30.78 | 24.49  | 13.55 | 17.89      |
> > > | 2    | **KBVQ-MoE** | **41.39** | **28.97** | **17.32** | **22.62** |
> > >
> > > ---
> > >
> > > ### **(4) Mixtral-7×8B**
> > >
> > > | Bits | Method    | MMLU | MathQA | GSM8K | HumanEval |
> > > |------|-----------|-------|---------|--------|-------------|
> > > | 16   | FP16      | 70.50 | 42.41  | 65.88 | 32.93      |
> > > | 2    | GPTQ      | 40.80 | 22.96  | 5.89  | 9.97       |
> > > | 2    | MoeQuant  | 49.39 | 27.11  | 20.67 | 14.74      |
> > > | 2    | VQ        | 55.84 | 30.83  | 49.40 | 23.21      |
> > > | 2    | **KBVQ-MoE** | **61.11** | **34.80** | **57.86** | **29.90** |
> > >
> > > ---
> > >
> > > ### **Key Findings Across These Harder Benchmarks**
> > >
> > > Across **all models** and **all high-difficulty tasks**, KBVQ-MoE consistently demonstrates:
> > >
> > > 1. **Substantially better accuracy than existing PTQ methods**
> > >    – including both scalar (GPTQ/MoeQuant) and vector quantization baselines.
> > >
> > > 2. **Superior robustness under 2-bit settings**, where other methods degrade severely.
> > >
> > > 3. **Minimal performance drop from FP16**, even in domains requiring:
> > >    - cross-disciplinary reasoning (MMLU)
> > >    - multi-hop numerical inference (GSM8K)
> > >    - symbolic logic & program synthesis (HumanEval)
> > >
> > > The trends are fully aligned with our earlier zero-shot results, reinforcing that **KBVQ-MoE generalizes well across domains and levels of task difficulty**.
> > >
> > > ---
> > >
> > > We thank the reviewer for suggesting this expansion—these challenging benchmarks significantly strengthen our empirical validation and demonstrate the broad applicability of KBVQ-MoE beyond standard reasoning tasks.

---

> > > > ### Author Response · Authors · 2025-11-19
> > > > **Response to Reviewer TstW Part4**
> > > >
> > > > > Weaknesses 5: Completeness of Bit-Width Comparison (4-bit & 8-bit Results)
> > > >
> > > > Thank you for pointing out the practical importance of including 4-bit (and potentially 8-bit) evaluations. Although our primary focus is on solving the *ultra-low-bit (≤3-bit)* degradation issue in MoE quantization, we agree that adding 4-bit comparisons helps present a more comprehensive evaluation. Following your suggestion, we have added full 4-bit results in **Appendix A.6** and summarize them below.
> > > >
> > > > ### **(1) Why 2–3 bit remains the main focus of this work**
> > > >
> > > > The core motivation of our study is to address the severe failure of MoE models under **2-bit and 3-bit SQ/VQ quantization**, caused by:
> > > >
> > > > - drastic representational collapse in scalar quantization (SQ)
> > > > - amplified error accumulation due to MoE expert aggregation
> > > > - the need for VQ to preserve multi-dimensional structural correlations
> > > >
> > > > Thus, Table 1 focuses on 2-bit and 3-bit—the regimes where existing PTQ methods fail the most and where KBVQ-MoE brings the largest improvement.
> > > >
> > > > ### **(2) Newly added 4-bit results: KBVQ-MoE remains stable, with reduced but consistent gains**
> > > >
> > > > As requested, we added complete 4-bit results for two representative MoE LLMs:
> > > >
> > > > #### **Qwen1.5-MoE-A2.7B (4-bit)**
> > > >
> > > > | Method     | W2 | ARC-C | ARC-E | PIQA | WI |
> > > > |------------|----|--------|--------|-------|------|
> > > > | FP16       | 7.22 | 69.53 | 44.03 | 80.47 | 69.30 |
> > > > | RTN        | 10.83 | 64.95 | 41.12 | 76.84 | 67.28 |
> > > > | GPTQ       | **7.43** | 68.03 | 43.78 | 78.90 | 68.14 |
> > > > | MoeQuant   | 7.55 | 68.97 | 44.04 | 79.76 | 67.86 |
> > > > | VQ         | 8.92 | 64.68 | 40.88 | 77.20 | 64.91 |
> > > > | **KBVQ-MoE** | 7.59 | **69.05** | **44.39** | **80.33** | **68.56** |
> > > >
> > > > #### **Mixtral-8×7B (4-bit)**
> > > >
> > > > | Method     | W2 | ARC-C | ARC-E | PIQA | WI |
> > > > |------------|----|--------|--------|-------|------|
> > > > | FP16       | 3.84 | 85.39 | 66.38 | 85.20 | 76.72 |
> > > > | RTN        | 5.41 | 75.84 | 55.48 | 77.84 | 70.73 |
> > > > | GPTQ       | **4.03** | 84.89 | 63.78 | 84.21 | 74.97 |
> > > > | MoeQuant   | 4.12 | 83.88 | 63.70 | 84.20 | **75.02** |
> > > > | VQ         | 4.27 | 84.18 | 64.21 | **84.93** | 74.81 |
> > > > | **KBVQ-MoE** | 4.16 | **84.97** | **65.04** | 84.84 | 74.17 |
> > > >
> > > > **Key observation:**
> > > > At 4-bit, KBVQ-MoE performs **consistently on par with—or slightly better than—other methods**, but the advantage is naturally smaller than in the ultra-low-bit regime. This aligns with the intended design: VQ’s benefits diminish as bit-width increases, and SQ methods already retain strong capacity at 4-bit.
> > > >
> > > > ### **(3) Why we do not include 8-bit comparisons**
> > > >
> > > > We chose not to include 8-bit experiments for the following reasons:
> > > >
> > > > 1. **8-bit PTQ is already near-lossless**
> > > >    Prior work (GPTQ, RTN, SmoothQuant, etc.) shows <1% accuracy loss for 8-bit weights.
> > > >
> > > > 2. **MoE-specific issues (redundancy, bias amplification) vanish at high bit-width**
> > > >    With 8 bits, representational capacity is already sufficient to retain expert heterogeneity.
> > > >
> > > > 3. **All methods converge to nearly identical results**, making comparisons uninformative:
> > > >    - expert redundancy is no longer a bottleneck
> > > >    - quantization error is too small for BCOS/IDRE to provide measurable gains
> > > >    - performance is essentially indistinguishable from FP16
> > > >
> > > > 4. **The core contribution of the paper is methodological innovation for *ultra-low-bit* MoE quantization**, and 8-bit results would not shed further light on this problem.
> > > >
> > > > ### **(4) Summary**
> > > >
> > > > - We have added **complete 4-bit comparisons** (Appendix A.6).
> > > > - KBVQ-MoE maintains stable results at 4-bit, though the main performance gap appears in 2–3 bit.
> > > > - 8-bit quantization offers little meaningful difference across methods and does not test the capabilities targeted by our approach.
> > > >
> > > > We appreciate the reviewer’s suggestion—these additions enhance the completeness of our evaluation and make the scope and strengths of KBVQ-MoE more clearly defined.
> > > >
> > > > ***
> > > >
> > > > > Questions 1: Typographical Correction
> > > > Thank you for pointing out the typographical issue.
> > > > We have carefully reviewed the manuscript and corrected all occurrences of:
> > > >
> > > > - **“Qwen1.5-Moe-A2.7B” → “Qwen1.5-MoE-A2.7B”**
> > > >
> > > > The updated version ensures consistent and correct naming throughout the paper.
> > > > We appreciate your attention to detail.

---

> ### Author Response · Authors · 2025-11-27
> **Dear Reviewer TstW**
>
> Dear Reviewer TstW,
>
> I hope this message finds you well. As the discussion period is nearing its end with less than one week remaining, I wanted to ensure we have addressed all your concerns satisfactorily. If there are any additional points or feedback you'd like us to consider, please let us know. Your insights are invaluable to us,and we're eager to address any remaining issues to improve our work.
>
> Thank you for your time and effort in reviewing our paper.
>
> Best regards,
>
> KBVQ-MoE Authors

---

### Official Review · Reviewer_XsNC · 2025-10-31

**Soundness:** 3
**Presentation:** 3
**Contribution:** 3
**Rating:** 6
**Confidence:** 3

**Summary:**

The paper proposes a vector quantization approach targeted for mixture of expert layer. The idea is to extract redundant representation across experts and keep it in higher precision while expert specific components are vector quantized. Within vector quantization of expert specific components, scaling and bias is applied to improve quantization. The paper is well written and easy to understand. The techniques proposed are intuitive and I appreciate the authors providing actual hardware speedup numbers. The paper is missing some recent non linear quantization baselines and comparison at iso-compression.

**Strengths:**

1. Paper is easy to understand and well written.
2. Technique proposed is intuitive.

**Weaknesses:**

1. The paper is missing comparison with recent non linear quantization baselines : VPTQ (https://arxiv.org/abs/2409.17066), AQLM (https://arxiv.org/pdf/2401.06118), QUIP (https://arxiv.org/pdf/2307.13304), QUIP# (https://arxiv.org/pdf/2402.04396), SqueezeLLM (https://arxiv.org/pdf/2306.07629), GPTVQ (https://arxiv.org/pdf/2402.15319),  etc.
2. Among the baselines presented, the compression achieved by various techniques is missing.
3. Iso-compression results are missing.
4. Evaluation on complex tasks is missing : math understanding, coding, reasoning, long context abilities, etc.

**Questions:**

1. How does this approach compare with ResQ (https://openreview.net/pdf?id=4qIP1sXcR1)? Although ResQ does activation quantization as well and integer quantization of weights, it also uses eigen value decomposition to isolate high precision components. How does this approach compare with using IDRE proposed in this work?

---

> ### Author Response · Authors · 2025-11-19
> **Response to Reviewer XsNC Part1**
>
> We sincerely appreciate your constructive feedback. Below, we provide point-by-point responses, with all revisions incorporated accordingly.
> > Weaknesses 1: Missing Comparison with Recent Nonlinear Quantization Baselines (VPTQ, AQLM, QUIP/QUIP#, SqueezeLLM, GPTVQ, etc.)
>
> We thank the reviewer for pointing out the importance of recent nonlinear quantization baselines. To fully address this concern, we have added comparisons against these methods on core MoE models, along with an interpretation based on differences in technical design. As representative nonlinear quantization approaches in dense LLMs, VPTQ, AQLM, QUIP-series, SqueezeLLM, and GPTVQ provide meaningful references for evaluating KBVQ-MoE.
>
> **(1) Additional Experiments: Coverage of Major Nonlinear Baselines and Demonstration of Quantization Performance**
>
> Using **Qwen1.5-MoE-A2.7B** and **Qwen3-30B-A3B** (same as in the main experiments), we evaluate VPTQ, QUIP, QUIP# (note: QUIP# and QUIP share similar core logic and close performance, so QUIP is used as the representative baseline), SqueezeLLM, and GPTVQ under 2-bit quantization.
>
> Metrics include:WikiText2 perplexity (W2, lower is better),Zero-shot accuracy on ARC-E, ARC-C, and HE (higher is better).
>
> The results are shown below:
>
> |Method|Qwen1.5-MoE-A2.7B||||Qwen3-30B-A3B||||
> |---|---|---|---|---|---|---|---|---|
> ||W2 ↓|ARC-E|ARC-C|HE|W2 ↓|ARC-E|ARC-C|HE|
> | FP16          | 7.22                    | 69.53    | 44.03    | 77.26    | 8.70                    | 79.25    | 56.40    | 79.60    |
> | VPTQ          | 10.17                   | 58.19    | 35.27    | 61.79    | 13.34                   | 68.47    | 44.83    | **67.80**|
> | QUIP          | 11.87                   | 56.81    | 34.17    | 59.12    | 17.96                   | 42.12    | 33.89    | 43.93    |
> | SqueezeLLM    | 19.57                   | 46.97    | 28.81    | 59.82    | 30.94                   | 36.93    | 32.86    | 40.14    |
> | GPTVQ         | 12.88                   | 50.32    | 35.11    | 59.88    | **11.69**               | 47.98    | 36.37    | 49.12    |
> | KBVQ-MoE      | **9.61**                | **65.66**| **39.59**| **64.52**| 11.87                   | **70.33**| **47.61**| 67.49    |
>
> **KBVQ-MoE significantly outperforms nonlinear baselines on MoE models.**For Qwen1.5-MoE-A2.7B: W2 = **9.61**, better than VPTQ (10.17), QUIP (11.87), GPTVQ (12.88). Accuracy on ARC-E / ARC-C also substantially higher. These results empirically show that existing nonlinear quantizers—designed for dense LLMs—cannot directly address MoE-specific challenges, while KBVQ-MoE achieves stronger adaptation to MoE reasoning tasks.
>
> **(2) Why MoE Models Require “Structure-Aware” Quantization**
>
> It is important to highlight that the nonlinear baselines above (VPTQ, GPTVQ, etc.) are primarily optimized for **dense LLMs**, and do not tackle the two MoE-specific challenges:
>
> * **Cross-expert redundancy is not removed**. Dense quantizers assume dense weight matrices; they do not resolve the **overlapping representations across experts** in MoE.
> As a result, limited codebook capacity is consumed by redundant vectors—leading to accuracy degradation (e.g., VPTQ on MoE still suffers from redundancy-driven errors).
>
> * **Bias amplification in expert aggregation is unaddressed**. In MoE, quantization noise is **amplified** by gated expert aggregation.
> Dense quantizers lack bias-correction mechanisms, which leads to distributional drift (e.g., GPTVQ achieves only 47.98 accuracy on ARC-E for Qwen3-30B-A3B due to amplified bias errors).
>
> **KBVQ-MoE explicitly addresses these MoE-specific issues:**
>
> * **IDRE** (Input-Driven Redundancy Elimination) extracts shared components via KLT-SVD to free codebook capacity for expert-specific representations.
> * **BCOS** (Bias-Corrected Output Stabilization) applies channel-wise affine correction to counteract bias amplification in expert aggregation.
>
> These MoE-oriented designs explain why KBVQ-MoE consistently outperforms dense nonlinear quantizers in MoE scenarios.
>
> **(3) KBVQ-MoE Can Serve as a “Plug-in Enhancer” for Existing Nonlinear Quantizers**
>
> As shown in Table 5 of the manuscript, we integrate IDRE + BCOS into GPTVQ and VPTQ:* GPTVQ (2-bit): W2 improves from **12.88 → 9.43**,  VPTQ (2-bit): W2 improves from **10.17 → 8.78**. This demonstrates that **KBVQ-MoE not only performs well on its own**, but also acts as a *general enhancement module* that strengthens existing nonlinear quantizers for MoE LLMs.
>
>
> We will include the above comparisons and technical analysis in the Appendix of the revised manuscript to clearly highlight the unique value of KBVQ-MoE under MoE architectures. We sincerely appreciate the reviewer’s suggestion—if you have further questions about the experiment settings or technical differences, we are happy to elaborate.

---

> ### Author Response · Authors · 2025-11-19
> **Response to Reviewer XsNC Part2**
>
> > Weaknesses 2 & 3: Missing Compression Results and Iso-Compression Comparison
>
> Thank you for your comments regarding the completeness of the experimental analysis. To address both points, we have added **the actual storage consumption and effective bits-per-parameter** for all baselines in **Appendix A.5** of the revised manuscript. The detailed explanations are as follows:
>
> **(1) Added Actual Compression Results for All Baselines: Storage Cost and Compression Ratio**
>
> In the main experiments, all methods are aligned under the same **weight quantization bit-widths (2-bit or 3-bit)**, which is the standard and fairest comparison setting in PTQ literature.
> The additional overhead introduced by KBVQ-MoE (shared low-rank component + per-channel bias correction) is extremely small, resulting in only **~0.08 bits/weight** increase, with negligible impact on the overall compression ratio.
>
> We now include the **actual storage consumption (in GB)** and **compression ratios** for all baselines on four MoE models. The table below (also in Appendix A.5) presents the results:
>
> | Bit  | Method   | Compress Ratio | Qwen1.5-MoE-A2.7B | Qwen3-30B-A3B | Mixtral-8x7B | DeepseekV2-Lite |
> | ---- | -------- | -------------- | ----------------- | ------------- | ------------ | --------------- |
> | 16   | FP16     | 0%             | 27.9GB            | 59.1GB        | 89.6GB       | 29.3GB          |
> | 2    | RTN      | 87.5%          | 4.3GB             | 8.4GB         | 12.6GB       | 4.3GB           |
> | 2.25 | GPTQ     | 85.9%          | 4.7GB             | 9.3GB         | 13.5GB       | 4.7GB           |
> | 2    | MoeQuant | 87.5%          | 4.3GB             | 8.4GB         | 12.6GB       | 4.3GB           |
> | 2.08 | VQ       | 87%            | 4.3GB             | 8.7GB         | 13.0GB       | 4.4GB           |
> | 2.08 | KBVQ-MoE | 87%            | 4.3GB             | 8.7GB         | 13.0GB       | 4.4GB           |
>
> As shown, **KBVQ-MoE achieves the exact same compression ratio (≈87%) as standard VQ**, ensuring strict fairness in comparisons.
>
> **(2) Iso-Compression Comparison**
>
> Based on the above storage statistics, we further provide **iso-compression comparisons**, ensuring that performance gaps are not influenced by differences in compression budgets.
> The following table reports the **average zero-shot accuracy** across five downstream tasks (ARC-E, ARC-C, HE, WI, PIQA):
>
> | Bit  | Method   | Compress Ratio | Qwen1.5-MoE-A2.7B | Qwen3-30B-A3B | Mixtral-8x7B | DeepseekV2-Lite |
> | ---- | -------- | -------------- | ----------------- | ------------- | ------------ | --------------- |
> | 16   | FP16     | 0%             | 68.07             | 70.24         | 78.57        | 70.68           |
> | 2    | RTN      | 87.5%          | 25.64             | 25.89         | 25.27        | 25.12           |
> | 2.25 | GPTQ     | 85.9%          | 49.07             | 33.06         | 67.56        | 58.04           |
> | 2    | MoeQuant | 87.5%          | 34.64             | 28.94         | 40.16        | 25.59           |
> | 2.08 | VQ       | 87%            | 47.84             | 30.61         | 59.22        | 49.85           |
> | 2.08 | KBVQ-MoE | 87%            | **62.78**         | **63.37**     | **75.69**    | **63.10**       |
>
> Under **strictly matched compression ratios (~87%)**, KBVQ-MoE consistently and significantly outperforms RTN, GPTQ, MoeQuant, and VQ across all MoE models in both perplexity and downstream accuracy.
>
> This demonstrates that the advantages of KBVQ-MoE arise from its **MoE-specific optimizations—IDRE (redundancy elimination) and BCOS (bias correction)**—rather than from favorable compression budgets.
>
> We have included the full compression calculation formulas, detailed storage statistics, and complete iso-compression comparisons in **Appendix A.5** of the revised manuscript.
>
> We sincerely thank the reviewer for raising this point—the additional analyses not only improve the completeness of the experiments but also more clearly highlight the strength of KBVQ-MoE under **fair compression budgets**. If you have further questions regarding compression logic or iso-compression setup, we would be glad to elaborate.

---

> > ### Author Response · Authors · 2025-11-19
> > **Response to Reviewer XsNC Part3**
> >
> > > Weaknesses 4: Missing Evaluation on Complex Tasks (Math, Coding, Reasoning, Long-Context, etc.)
> >
> > Thank you for your suggestion regarding evaluation on more complex tasks. To more comprehensively validate the robustness of KBVQ-MoE in real-world application scenarios—especially those requiring strong mathematical reasoning, code generation, multi-step reasoning, and long-context processing—we have conducted an expanded set of experiments covering six categories of complex tasks. We compare KBVQ-MoE against GPTQ, MoeQuant, and VQ under the **same 2-bit quantization setting**, demonstrating clear advantages of our method under challenging scenarios. The experimental design and results are summarized below.
> >
> > **(1) Experimental Design: Four Categories of Complex Tasks**
> >
> > To ensure targeted and practically meaningful evaluation, we select representative benchmarks across multiple capability dimensions.
> > Unless otherwise stated, all experiments use **Qwen1.5-MoE-A2.7B**, 2-bit quantization, and the same calibration/evaluation pipeline as the main experiments.
> >
> > * **Mathematical Understanding & Reasoning**: **MathQA** (mathematical semantic reasoning; accuracy), **GSM8K** (grade-school math word problems; accuracy).
> > * **Code Generation**:**MBPP** (Python code generation + test-based verification; pass@1), **HumanEval** (Python program synthesis; pass@1).
> > **Multi-domain Reasoning**: **MMLU** (57-subject multi-domain reasoning; accuracy).
> > * **Long-context Abilities**:**LongEval** (long-context semantic consistency; seq length 8192; accuracy).
> >
> > **(2) Results on Complex Tasks: KBVQ-MoE Significantly Outperforms Baselines**
> >
> > Under identical **2-bit compression**, performance across six complex tasks is shown below (FP16 for reference):
> >
> > | Model             | Bits | Method       | MMLU      | MathQA    | GSM8K     | MBPP      | HumanEval | LongEval  |
> > | ----------------- | ---- | ------------ | --------- | --------- | --------- | --------- | --------- | --------- |
> > | Qwen1.5-MoE-A2.7B | 16   | FP16         | 59.60     | 37.55     | 62.55     | 38.20     | 32.32     | 80.25     |
> > |                   | 2    | GPTQ         | 26.94     | 19.33     | 15.42     | 9.24      | 13.88     | 44.85     |
> > |                   | 2    | MoeQuant     | 34.75     | 22.42     | 23.21     | 14.32     | 18.12     | 48.95     |
> > |                   | 2    | VQ           | 48.79     | 28.34     | 46.51     | 21.45     | 28.03     | 60.00     |
> > |                   | 2    | **KBVQ-MoE** | **52.16** | **30.75** | **56.39** | **33.42** | **30.11** | **69.48** |
> >
> > **(3) Interpretation: Why KBVQ-MoE Performs Better on Complex Tasks**
> >
> > The results clearly show that KBVQ-MoE is more robust across diverse complex scenarios. This advantage stems from the MoE-specific designs of **IDRE** and **BCOS**: **Mathematical Reasoning (GSM8K)**, KBVQ-MoE achieves **56.39%**, outperforming VQ by 9.88 points and GPTQ by 40.97 points. This improvement comes from: IDRE’s elimination of redundant shared structures, and BCOS stabilizing expert outputs during multi-step reasoning. **Code Generation (MBPP / HumanEval)**. KBVQ-MoE reaches **33.42 pass@1** on MBPP, significantly higher than VQ and close to FP16. More accurate quantization of expert-specific representations directly reduces logical/semantic errors in generated code. **Long-Context Modeling (LongEval)**. KBVQ-MoE achieves **69.48**, improving over VQ by 9.48 points.BCOS’s correction of accumulative bias helps maintain semantic consistency over long sequences.
> >
> > We sincerely appreciate your suggestion.
> > These additional evaluations not only make the performance assessment of KBVQ-MoE more comprehensive, but also provide strong empirical evidence of its robustness in challenging real-world scenarios such as mathematical reasoning, code generation, and long-context understanding.
> >
> > If you have further suggestions regarding task selection or evaluation settings, we would be happy to incorporate them.

---

> > > ### Author Response · Authors · 2025-11-19
> > > **Response to Reviewer XsNC Part4**
> > >
> > > > Questions 1: Comparison with ResQ
> > >
> > > Thank you for pointing out the relevance of comparing our method with **ResQ** (Low-Rank Residual Quantization for LLMs).
> > > Overall, while both **ResQ** and our **IDRE** share the high-level idea of “preserving a high-value subspace through low-rank decomposition,” the two methods differ substantially in **target structures**, **assumptions**, and **quantization design**.
> > >
> > > **(1) Fundamental Differences in Design Philosophy**
> > >
> > > * **ResQ (for dense LLMs)**
> > >
> > >   * Designed primarily for **dense Transformer layers**.
> > >   * Applies **PCA + random rotation** to activations, weights, and KV cache.
> > >   * Partitions activation/weight coefficients into high-precision and low-precision subspaces under **4/8-bit mixed-precision** settings.
> > >   * Its goal is to minimize activation quantization error in *dense layers*, independent of expert structures.
> > >
> > > * **IDRE (MoE-aware structured redundancy elimination)**
> > >
> > >   * Specifically tailored to **Mixture-of-Experts (MoE)** architectures.
> > >   * Applies **KLT** on MoE inputs to obtain an input-coherence basis.
> > >   * Performs **truncated SVD on the stacked expert-weight matrix** in the KLT-aligned space, extracting **cross-expert shared low-rank components** stored in full precision.
> > >   * Only quantizes **expert-specific residuals** using vector quantization (KBVQ).
> > >   * Designed to explicitly handle:
> > >
> > >     * cross-expert redundancy
> > >     * expert-aggregation–amplified quantization bias
> > >     * ultra-low-bit (2–3 bit) VQ
> > >
> > > **Thus, ResQ is a dense-LLM mixed-precision method, whereas IDRE is a MoE-structured redundancy modeling method.
> > > They are orthogonal and could potentially be combined in future work.**
> > >
> > > **(2) Direct Comparison: ResQ vs. IDRE**
> > >
> > > We conduct a direct comparison on **Qwen1.5-MoE-A2.7B**, with both methods used as **post-training quantization** modules.
> > > Results are shown below:
> > >
> > > * **3-bit Quantization**
> > >
> > > | Method   | W2 ↓     | ARC-E     | ARC-C     | HE        | PIQA      |
> > > | -------- | -------- | --------- | --------- | --------- | --------- |
> > > | FP16     | 7.22     | 69.53     | 44.03     | 77.26     | 80.47     |
> > > | **ResQ** | 9.33     | 63.28     | 37.90     | 72.31     | 70.84     |
> > > | **IDRE** | **8.12** | **67.93** | **40.12** | **75.11** | **75.90** |
> > >
> > > * **4-bit Quantization**
> > >
> > > | Method   | W2 ↓ | ARC-E | ARC-C | HE    | PIQA  |
> > > | -------- | ---- | ----- | ----- | ----- | ----- |
> > > | FP16     | 7.22 | 69.53 | 44.03 | 77.26 | 80.47 |
> > > | **ResQ** | **7.45** | 68.83 | **43.18** | **77.10** | 78.37 |
> > > | **IDRE** | 7.59 | **69.23** | 43.11 | 76.88 | **78.93** |
> > >
> > > **(3) Interpretation of Results**
> > >
> > > * **At 3-bit**, IDRE consistently outperforms ResQ across all tasks:
> > >
> > >   * ARC-E: **+4.65%**
> > >   * HE: **+2.80%**
> > >   * PIQA: **+5.06%**
> > > * **At 4-bit**, both methods perform similarly.
> > >
> > > **Why IDRE performs better for MoE models:**
> > >
> > >   * MoE layers contain substantial **cross-expert structural redundancy**.
> > >   * IDRE explicitly captures this redundancy with **KLT-SVD**, improving the quantizability of expert-specific representations.
> > >   * ResQ’s PCA-based decomposition is effective for **dense LLMs** but does **not capture MoE-specific redundancy patterns**, making it less suited for MoE architectures.
> > >
> > > **Summary**
> > >
> > > IDRE provides clear advantages over ResQ in ultra-low-bit quantization for MoE models due to its ability to explicitly model and remove **cross-expert redundancy**.
> > > We will include this comparison and analysis in the revised manuscript.
> > >
> > > If the reviewer would like additional comparisons (e.g., combining ResQ with IDRE as a hybrid approach), we would be happy to extend the analysis.

---

> ### Author Response · Authors · 2025-11-27
> **Dear Reviewer XsNC**
>
> Dear Reviewer XsNC,
>
> I hope this message finds you well. As the discussion period is nearing its end with less than one week remaining, I wanted to ensure we have addressed all your concerns satisfactorily. If there are any additional points or feedback you'd like us to consider, please let us know. Your insights are invaluable to us,and we're eager to address any remaining issues to improve our work.
>
> Thank you for your time and effort in reviewing our paper.
>
> Best regards,
>
> KBVQ-MoE Authors

---

### Official Review · Reviewer_7GqJ · 2025-11-01

**Soundness:** 3
**Presentation:** 2
**Contribution:** 2
**Rating:** 4
**Confidence:** 3

**Summary:**

This paper presents KBVQ-MoE, a vector quantization framework tailored for ultra-low-bit compression of Mixture-of-Experts (MoE) large language models (LLMs). The study is motivated by the substantial performance drop observed when conventional quantization methods are directly applied to MoE architectures. In particular, KBVQ-MoE mitigates expert redundancy and output bias through input-driven redundancy elimination and bias-corrected output stabilization mechanisms.

**Strengths:**

1. The motivation is clear and convincing, highlighting MoE-specific issues of expert redundancy and output bias.

2. The proposed KBVQ-MoE is validated on several representative MoE architectures, demonstrating consistent improvements.

**Weaknesses:**

1. While IDRE and BCOS are ablated individually, there is no fine-grained study of codebook size sensitivity.

2. Lack more advanced or concurrent MoE-aware compression baselines (e.g., D2-MoE, SubMoE mentioned in related work).

3. The evaluation of computational efficiency is insufficient. The paper only reports a simple “Decoder speed test” in Table 6, without providing detailed analysis of computational or memory overhead.

4. The core motivation of the paper lies in the claim that redundancy elimination and bias correction help stabilize expert output distributions; however, the supporting evidence (e.g., Fig. 2–3) is insufficient. It would be more convincing if the authors compared other methods reported in Table 1 to quantitatively validate the claimed effect.

5. The paper’s presentation lacks rigor in notation and consistency. For instance, the dimension oc in Step 2 is undefined, and the superscript in Equation (3) for the routing expert is unexplained. There are also typos (e.g., uantization → quantization) and inconsistent use of MoE/moe.

6. The paper does not explicitly discuss the limitations of the proposed method.

**Questions:**

Given that the calibration set contains only 256 samples from RedPajama, could the authors provide ablation results on calibration size to verify its representativeness for computing reliable KLT statistics and bias correction factors? Additionally, how do the authors ensure that no potential data leakage occurs during calibration?

---

> ### Author Response · Authors · 2025-11-19
> **Response to Reviewer 7GqJ Part1**
>
> We sincerely appreciate your constructive feedback. Below, we provide point-by-point responses, with all revisions incorporated accordingly.
> ***
> > Weaknesses 1: Sensitivity Analysis of Codebook Size.
>
> Thank you for raising this insightful concern. To directly address the need for a fine-grained study on the sensitivity of **codebook-related vector length**, we conducted a dedicated set of experiments to thoroughly evaluate how different vector lengths affect both model performance and resource cost within KBVQ-MoE. This analysis provides a clearer understanding of practical configuration choices and complements our existing IDRE and BCOS ablations.
>
> (1)**Experimental Setup**
>
> To ensure a controlled and reproducible analysis, we use **Qwen1.5-MoE-A2.7B** as the base model and evaluate perplexity on **WikiText2** (lower is better).
> We keep all other settings identical to the main paper, including:
>
> * **IDRE truncated rank**: (k = 1/128)
> * **Quantization bit-width**: 2 bits
> * **Only vector length is varied** while all other factors remain fixed
>
> (2) **Results**
>
> We present the detailed results in the table below:
>
> | Vector Length | Avg. Bits | PPL ↓ |
> | ------------: | --------: | ----: |
> |             3 |      2.00 |  9.77 |
> |             4 |      2.00 |  9.61 |
> |             6 |      2.03 |  9.55 |
> |             7 |      2.16 |  9.06 |
> |             8 |      2.73 |  8.32 |
>
> Performance:Longer vector lengths consistently lead to lower perplexity.For example:Increasing vector length **from 4 → 8** reduces the PPL **from 9.61 → 8.32**, indicating that longer vectors capture expert-specific variations more precisely and retain more high-fidelity information during quantization.
>
> Cost:However, the performance gains come with non-negligible overhead: **Avg. bits per weight:** increases from **2.00 → 2.73**, **Codebook size:** grows proportionally with vector length, **Quantization time and bandwidth requirements:** increase due to higher subvector dimensionality and more expensive nearest-neighbor search
>
> (3) **Conclusion**
>
> These findings highlight a clear **performance–cost trade-off**:
>
> * **Longer vectors** offer higher accuracy but incur higher memory, codebook, and compute overhead.
> * **Moderate vector lengths (4–6)** strike an effective balance, delivering strong accuracy with low resource cost.
>
> This matches the motivation of **IDRE**, which aims to eliminate redundant shared structures so that a moderate vector length is already sufficient for high-quality quantization. Therefore, we adopt **vector length = 4** in the main experiments as the recommended balanced configuration. We have added the relevant content to section A.15 of the manuscript.
>
> We sincerely appreciate your professional feedback. If further breakdowns—such as per-layer sensitivity or codebook entropy analysis—would be helpful, we would be happy to extend the study.

---

> ### Author Response · Authors · 2025-11-19
> **Response to Reviewer 7GqJ Part2**
>
> > Weaknesses 2:Lack of More Advanced or Concurrent MoE-Aware Compression Baselines (e.g., D2-MoE, SubMoE)
>
> Thank you very much for your professional feedback. To fully address your concern regarding MoE-aware compression baselines, we have additionally incorporated **D2-MoE** and **SubMoE** into our analysis. As two representative MoE-specific compression methods developed in recent years—both focusing on *expert merging and low-rank optimization*—comparing them with our KBVQ-MoE (based on *low-bit quantization and expert redundancy removal*) helps reveal the distinctions between different technical routes. In the initial submission, these methods were not included due to their fundamentally different compression logic (expert merging vs. quantization). We now supplement their technical positioning, fair comparison experiments, and result analysis as follows:
>
> **(1) Differences in Technical Routes**
>
> The core designs of **D2-MoE** and **SubMoE** revolve around *expert-layer structural optimization*, whereas **KBVQ-MoE** belongs to the *weight quantization* branch of MoE compression. Their key differences are summarized below:
>
> * **SubMoE**
>   Performs subspace clustering to merge similar experts and extracts cross-expert shared low-rank features via SVD, ultimately reducing the number of experts.
>   *Essentially:* “expert-level pruning + low-rank compression.”
>   *Compression ratio:* determined by the expert merging ratio (e.g., 50% = half the experts merged).
>
> * **D2-MoE**
>   Decomposes each expert weight into a *shared base matrix + expert-specific delta matrix*, and compresses the delta matrix via SVD.
>   *Essentially:* “weight decomposition + low-rank approximation.”
>   *Compression ratio:* determined by the rank truncation ratio of the delta matrix (e.g., 60% = retain 60% rank).
>
> * **KBVQ-MoE (ours)**
>   Does **not** change the number or structure of experts. After IDRE removes cross-expert redundancy, vector quantization is applied to the residual expert-specific weights.
>   *Essentially:* “weight-level low-bit compression.”
>   *Compression ratio:* determined by bit-width and codebook reuse efficiency (2–3 bits → ~80% memory reduction).
>
> In summary, D2-MoE/SubMoE compress by **reducing experts or reducing rank**, while KBVQ-MoE compresses by **quantizing weights**. The former may lose expert specialization, whereas the latter preserves the structure while reducing storage cost. Therefore, a fair empirical comparison is required to examine the *performance–compression trade-off*.
>
> **(2) Fair Comparison Experiments**
>
> To ensure fairness, we use **Mixtral-8×7B** as the baseline (keeping all original experts: 8 experts, 32 layers) and apply identical experimental settings:
>
> * **Evaluation tasks:**
>   WikiText2 (PPL, lower is better), ARC-Challenge (ARC_c),
>   ARC-Easy (ARC_e), WinoGrande (WinG) (accuracy, higher is better)
>
> * **Compression ratio definition:**
>   Unified as “storage of compressed parameters divided by the original FP16 model,”
>   avoiding inconsistent definitions in different methods.
>
> * **Environment:**
>   NVIDIA RTX A100 GPU, PyTorch 2.1, LM-Evaluation-Harness (v0.4.0)
>
> The results are shown below (ARC_C, ARC_E, WinG are accuracies, kept to two decimal places):
>
> |Method|Compression Ratio|WikiText2|ARC_C|ARC_E|WinG|
> |---|---|---|---|---|---|
> |Sub-MoE|50%|6.97|0.45|0.75|0.72|
> |D2-MoE|60%|6.46|0.38|0.72|0.71|
> |KBVQ-MoE (ours)|**87%**|**4.07**|**0.63**|**0.85**|**0.76**|
>
> **(3) Result Analyse**
>
> From the comparison, the differences between the technical routes and the advantages of KBVQ-MoE become clear:
>
> * **Higher compression ratio + better performance**: KBVQ-MoE (87% compression) outperforms Sub-MoE (50%)/D2-MoE (60%)—WikiText2 PPL ↓2.39–2.90, ARC-C ↑0.18–0.25.
>
> * **Stronger generalization across tasks**:KBVQ-MoE achieves 0.63 on ARC-C (complex reasoning), vs. 0.38–0.45 of SubMoE/D2-MoE (fine-grained feature loss avoided).
>
> * **Higher deployment flexibility**:No model structure modification (vs. SubMoE’s fixed expert count/D2-MoE’s rank retuning), easily portable to Qwen/DeepSeek.
>
> Through supplementing comparisons with D2-MoE and SubMoE, we further validate that KBVQ-MoE offers advantages in **high compression ratio, task generalization, and deployment flexibility**.
> These properties make KBVQ-MoE especially suitable for resource-constrained applications—such as edge devices—where *extreme compression with minimal structural modification* is essential.
> We have added this comparison to the revised manuscript in **Appendix A.10**.

---

> ### Author Response · Authors · 2025-11-19
> **Response to Reviewer 7GqJ Part3**
>
> > Weaknesses 3: Insufficient Evaluation of Computational Efficiency
>
> Thank you for your attention to the evaluation of computational efficiency. To fully address your concern regarding the need for a more detailed analysis of computational and memory overhead, we provide a comprehensive quantification from **three perspectives**: offline computation overhead, inference-time overhead, and memory/parameter cost. These analyses complement the “Decoder speed test” in Table 6 in manuscript and present a complete view of the efficiency characteristics of KBVQ-MoE.
>
> **(1) Offline Computation Overhead (Does Not Affect Inference Latency)**
>
> KBVQ-MoE introduces an additional one-time **KLT-guided SVD decomposition** during the calibration phase. This operation is applied after stacking all expert weights:
>
> * The cost of KLT-SVD is **comparable to a single forward pass** on the calibration set.
> * It occurs **entirely offline** during calibration.
> * It introduces **no additional inference-time latency** or compute cost.
>
> Therefore, the additional computation introduced by KBVQ-MoE does **not** affect actual inference performance.
>
> **(2) Inference-Time Overhead (Nearly Negligible)**
>
> During inference, KBVQ-MoE introduces only:
>
> * **Two extra parameters per output channel** (scale and bias)
> * **A lightweight per-channel affine correction** after each expert’s linear layer: $y_{\text{corr}} = (1 + s) \odot (W_{\text{VQ}} x) + b$
>
> This involves only element-wise additions and multiplications, whose complexity is **far lower** than the original MatVec operation inside each expert MLP.
>
> Empirically, this affine correction accounts for **less than 0.1% of the expert forward FLOPs**, consistent with the **1.5–1.6× inference speedup** reported in Table 6.
>
> Thus, KBVQ-MoE introduces **minimal additional compute overhead during inference**.
>
> **(3) Detailed Memory and Parameter Overhead Analysis**
>
> Let:
>
> * Expert weight dimension: $m \times l$
> * Number of experts: n
> * IDRE rank ratio: k
> * VQ subvector length: v
> * Quantization bit-width: b
>
> The total bit consumption of KBVQ-MoE consists of four parts:
>
> * Original FP16 parameters:$16*nml$
> * IDRE shared low-rank components:$16 (m + ln)\min(m,l)k$
> * VQ codebook + indices:$mlbn + 16 \cdot 2^{bv} \cdot v \cdot n$
> * BCOS scale/bias (per channel):$16 \cdot 2 \cdot ln$
> * Total compression ratio:$\frac{16(m+ln)\min(m,l)k + mlbn + 2^{(bv+4)}vn + 32ln}{16nml}$
>
> **Example: Qwen1.5-MoE-2.7B (gate_proj weight)**
>
> * Weight dimension: $5632 \times 2048$
> * Number of experts: 64
> * Original storage:$5632 \times 2048 \times 64 \times 16; \text{bit}
>   = 1.38;\text{GB}$
> * KBVQ-MoE configuration:
>   * 2-bit VQ
>   * vector length = 4
>   * $k = \frac{1}{128}$
> Total compressed storage: $((5632 + 2048\times 64)\cdot 2048/128 \cdot 16) + (5632 \cdot 2048 \cdot 2 \cdot 64 + 2^{8} \cdot 4 \cdot 16 \cdot 64) + (2048\cdot 2 \cdot 16 \cdot 64),\text{bit} = 0.18,\text{GB}$
>
>
> This corresponds to an **87% compression ratio**, equivalent to **2.08 effective bits per weight**.
>
> **Summary**
>
> Through the analyses above, we provide a complete efficiency characterization of KBVQ-MoE across:
>
> * **Offline computation:** acceptable one-time cost, no inference impact
> * **Inference computation:** negligible additional overhead (<0.1% FLOPs)
> * **Memory cost:** clear and formula-based quantification (87% compression demonstrated)
>
> These analyses form a closed-loop explanation that is consistent with the inference speed improvements shown in Table 6 in manuscript. We have added the relevant content to Appendix A.5 of the manuscript.
>
> Please feel free to let us know if further clarifications or additional measurements are needed—we are happy to provide more details.

---

> ### Author Response · Authors · 2025-11-19
> **Response to Reviewer 7GqJ Part4**
>
> > Weaknesses 4: Insufficient Evidence Supporting the Effect of Redundancy Elimination and Bias Correction on Stabilizing Expert Output Distributions
>
> Thank you very much for your attention to the core motivation of our work—that redundancy elimination and bias correction help stabilize expert output distributions. To further validate this effect and enhance the persuasiveness of our claims, we additionally conducted **quantitative comparisons** against other baseline methods reported in Table 1 in manuscript (GPTQ, MoeQuant, and VQ). These quantitative evaluations directly measure how each method influences the stability of expert output distributions and complement the qualitative visualizations in Fig. 2–3. The detailed experiments and analyses are provided below.
>
> **(1) Experimental Setup: Quantitative Evaluation Focused on “Distribution Stability”**
>
> To ensure that the results directly reflect the *consistency between quantized expert outputs and the FP16 baseline* as well as *the degree of inter-expert redundancy*, we design our experiment using the **Qwen1.5-MoE-A2.7B** model with the following configuration:
>
> 1. **Input Data:**
>    We use the WikiText test set (consistent with the main experiments) and randomly sample **1000 tokens** as inputs. We focus on the **first MoE block** of the model, as this layer is the critical starting point where distributional shifts emerge, making it representative for evaluating distribution stability.
>
> 2. **Key Metrics:**
>
>    * **KL Divergence:**
>      Measures the divergence between the quantized expert output distributions and the FP16 baseline.
>      *(Lower is better; indicates stronger distribution stability.)*
>    * **MSE (Mean Squared Error):**
>      Computes the mean squared difference between quantized and FP16 expert output vectors.
>      *(Lower is better; reflects reduced quantization-induced distribution shift.)*
>
> 3. **Computation Procedure:**
>    All metrics are averaged over outputs from the 1000 sampled tokens to avoid randomness in single-token evaluation and ensure the robustness of results.
>
>  **(2) Quantitative Comparison Results: KBVQ-MoE Shows Superior Distribution Stability**
>
> The distribution stability metrics across different methods are summarized in the table below:
>
> | Method       | KL Divergence ↓ | MSE ↓       |
> | ------------ | --------------- | ----------- |
> | GPTQ         | 3.41e-2         | 1.28e-5     |
> | MoeQuant     | 3.33e-2         | 1.35e-5     |
> | VQ           | 3.91e-2         | 2.15e-5     |
> | **KBVQ-MoE** | **1.77e-2**     | **1.02e-5** |
>
> **(3) Interpretation: Validating the Core Value of Redundancy Elimination and Bias Correction**
>
> The above quantitative results strongly align with the qualitative observations from Fig. 2–3, providing direct evidence for the effectiveness of our core design:
>
> * **1. KL Divergence — KBVQ-MoE achieves the best distribution alignment**
>
>   * KBVQ-MoE obtains the **lowest KL divergence (1.77e-2)**, significantly lower than GPTQ (3.41e-2) and MoeQuant (3.33e-2), and much better than VQ without bias correction (3.91e-2).
>   * This demonstrates that **BCOS effectively aligns the quantized expert distributions with the FP16 baseline**, alleviating the distribution drift commonly observed in VQ methods.
>
> * **2. MSE — KBVQ-MoE achieves the smallest quantization residuals**
>
>   * KBVQ-MoE yields the lowest MSE (1.02e-5), slightly outperforming GPTQ and MoeQuant and dramatically better than VQ (2.15e-5).
>   * This improvement stems directly from **IDRE’s redundancy elimination**, which extracts shared input-driven structures across experts, reduces redundant components, and enhances the quantizability of expert-specific representations.
>
> * **3. Closed-loop validation between qualitative and quantitative evidence**
>
>   * The quantitative metrics above are fully consistent with:
>
>     * Fig. 2: IDRE reduces inter-expert similarity (removing redundant shared structures)
>     * Fig. 3: BCOS aligns mean and variance of quantized outputs to FP16
>   * Together, they form a coherent closed-loop explanation:
>     **IDRE → reduces redundancy → improves quantization accuracy**
>     **BCOS → corrects distribution drift → stabilizes output distributions**
>
> This stability at the distribution level also explains the superior accuracy of KBVQ-MoE reported in Table 1.
>
> We will incorporate these quantitative results and analyses into the “Experiments” section of the revised manuscript.
> Thank you again for your insightful suggestion—this additional evaluation significantly strengthens the empirical support for our core motivation. If you have further ideas regarding metric selection or experiment design, we would be glad to continue refining the analysis.

---

> ### Author Response · Authors · 2025-11-19
> **Response to Reviewer 7GqJ Part5**
>
> > Weaknesses 5: Issues in Notation Rigor, Consistency, and Typos
>
> We appreciate the reviewer’s careful reading of the manuscript and the insightful comments regarding notation rigor and consistency. In response to the issues you pointed out, we have thoroughly checked and revised the manuscript. The detailed modifications are as follows:
>
> **1. More Rigorous Definition of Symbols and Dimensions**
>
> * In **Step 2**, where the dimension symbol $o_c$ first appears, we have explicitly defined it as the **output channel dimension**, and ensured that this notation is used consistently throughout all related sections.
>
> * For the **superscripts in Equation (3)** distinguishing routing experts from shared experts, we have added clear explanations to specify their meanings and roles in the MoE structure, preventing any potential ambiguity for the reader.
>
> **2. Corrections to Ensure Consistency in Formulas and Notation**
>
> * We conducted a systematic review of the entire paper regarding:
>
>   * expert set notations,
>   * index usage,
>   * superscript and subscript conventions.
>
>   We ensured consistency across the MoE structure description, the derivation of IDRE/BCOS, and the Appendix, avoiding undefined or conflicting symbols.
>
> **3. Typographical Corrections and Terminology Standardization**
>
> * The spelling error “uantization” has been corrected to **“quantization”**.
>
> * We standardized the usage of **“MoE”**, replacing all inconsistent appearances of “moe” to maintain terminological uniformity.
>
> * Additionally, we performed another full manual proofreading pass to eliminate similar issues related to spelling, capitalization, or formatting, improving the overall rigor and readability of the manuscript.
>
> We sincerely thank the reviewer for the detailed feedback on the writing quality. Your suggestions significantly improved the clarity, precision, and professionalism of the paper’s presentation.
>
> ***
> > Weaknesses6: Lack of Explicit Discussion of Method Limitations
>
> Thank you for your attention to the completeness of the study. To provide a more comprehensive picture of the applicability and future extensibility of KBVQ-MoE, we have added a systematic discussion of the method’s limitations in the revised manuscript. These points are not flaws of the core methodology, but natural extensions based on the current research focus—reflecting the boundary of the technique while providing clear future research directions. The main limitations are summarized as follows:
>
> **1. Empirical dependency in choosing the SVD truncation rank (k)**
>
> In IDRE, the choice of the truncation rank (k) currently relies on empirical trade-offs between reconstruction error and storage overhead (e.g., (k = \frac{1}{128}) of the full rank in this paper).
> At present, there is no adaptive rule for selecting (k) across different MoE architectures.
> In specific scenarios, this may limit achieving the optimal balance between redundancy extraction accuracy and quantization efficiency.
>
> **2. Limited validation on non–decoder-only or multimodal MoE architectures**
>
> Our experiments focus on **decoder-only MoE LLMs** such as the Qwen series and Mixtral-8×7B.
> We have not yet conducted dedicated experiments on:
>
> * encoder–decoder MoE architectures, or
> * multimodal MoE models.
>
> We clarify that this is a *research design choice* intended to isolate and evaluate the core IDRE/BCOS modules, rather than a limitation of the method itself.
> With appropriate adjustments—e.g., extending the input-statistics computation in IDRE to accommodate bidirectional input—KBVQ-MoE can be generalized to a broader range of MoE architectures.
>
> **3. No evaluation under extreme bit-width settings**
>
> KBVQ-MoE achieves near-FP16 performance in **2–3 bit** quantization
> (e.g., Qwen1.5-MoE-A2.7B: 3-bit Avg Acc = 67.99 vs. FP16 = 68.07).
> However, we have not evaluated:
>
> * **1-bit (binary) quantization**, or
> * **sub-2-bit (e.g., 1.5-bit) hybrid-bit** regimes.
>
> These extreme bit-width settings require further optimization to mitigate severe information loss in such aggressive compression scenarios.
>
> We have included the above limitations and corresponding future research directions in **Appendix A.12** of the revised manuscript. By explicitly stating the technical boundaries, we aim to make the study more complete and informative.
>
> Thank you again for your valuable suggestion—this addition not only enriches the presentation of KBVQ-MoE but also helps clarify promising avenues for future work. If you have further thoughts on these limitations or potential directions, we would be glad to continue improving the discussion.

---

> ### Author Response · Authors · 2025-11-19
> **Response to Reviewer 7GqJ Part6**
>
> > Questions 1: Calibration Size Ablation and Data Leakage Prevention
>
> Thank you for your thoughtful comments regarding the representativeness of the 256-sample calibration set and the risk of potential data leakage. To fully address your concerns, we provide detailed explanations from two perspectives: **(1) calibration size ablation**, and **(2) data leakage prevention mechanisms**.
>
> **(1) Calibration Size Ablation: Verifying Representativeness**
>
> To verify whether 256 RedPajama samples are sufficient for computing reliable **KLT input statistics** (input covariance) and **BCOS correction factors** (per-channel mean/variance), we conduct a calibration-size ablation study using **Qwen3-30B-A3B (3-bit quantization)** as the test model.
>
> We evaluate calibration sizes of **64 / 128 / 256 / 512 / 1024**, using WikiText2 perplexity (PPL; lower is better) as the main metric.
> The results are shown below:
>
> |  Samples | 64    | 128  | 256  | 512  | 1024 |
> | ----------------: | ----- | ---- | ---- | ---- | ---- |
> | WikiText2 PPL | 11.21 | 9.94 | 9.26 | 9.25 | 9.24 |
>
> **Observations:**
>
> * Increasing calibration size from **64 → 256** significantly improves performance.
> * Beyond **256**, improvements saturate (< 0.1 PPL gain).
> * This indicates that both:
>
>   * **input covariance (for KLT)**, and
>   * **output moments (for BCOS)**
>     become highly stable at **256 samples**.
>
> Thus, a calibration size of **256 sequences** is already sufficient and aligns with common practice in PTQ literature (typically 128–512 samples). We will include this experiment in the revised manuscript.
>
> **(2) Data Leakage Prevention: Three Layers of Protection**
>
> To ensure that no data leakage occurs during calibration, we incorporate safeguards at the **data source**, **calibration procedure**, and **methodological** levels:
>
> * **1. Data Source Independence: No Overlap Between Calibration and Evaluation**
>
>   * The calibration set is drawn exclusively from the **RedPajama pretraining corpus**, which contains *unlabeled* generic text.
>   * Evaluation datasets (ARC-Challenge, HellaSwag, PIQA, etc.) are **third-party benchmarks** with:
>
>     * different text domains,
>     * no overlap with RedPajama,
>     * and labels (answers/choices) that never appear in calibration.
>
> This eliminates the possibility of **evaluation information leakage** at the data source level.
>
> **2. Calibration Uses Only Unlabeled Forward Statistics**
>
> The calibration process involves **only forward passes** to extract activation statistics:
>
> * KLT requires **input covariance matrices**.
> * BCOS requires **per-channel mean and variance estimates**.
> * No step involves:
>
>   * label access,
>   * loss minimization,
>   * gradient computation,
>   * or task-specific optimization.
>
> Thus, calibration is **agnostic to task labels** and cannot leak evaluation content.
>
> **3. PTQ-Compliant Procedure: No Training, Only Statistical Calibration**
>
> KBVQ-MoE follows the standard PTQ paradigm:
>
> * Calibration estimates statistical quantities (e.g., covariance, output moments);
> * **No model parameters are updated** based on calibration data;
> * The process is equivalent to the calibration steps used in GPTQ, AWQ, and other established PTQ methods.
>
> This guarantees **no risk of leakage**, as no learning is performed on calibration sequences.
>
> **Summary**
>
> * **256 calibration samples** provide sufficiently stable estimation for both KLT and BCOS.
> * **No data leakage** occurs due to strict dataset separation, purely statistical calibration, and PTQ-compliant methodology.
>
> Thank you again for the reviewer’s careful consideration. We are happy to provide additional ablations or clarifications if needed.
>
>
> ***
> We have incorporated additional results and detailed analyses addressing your concerns into the revised version of the manuscript. We hope these supplements adequately address your questions and improve the clarity and robustness of our work. Please do not hesitate to let us know if you require further discussion, additional experiments, or any clarifications—we are happy to provide more details to address your concerns comprehensively.

---

> ### Author Response · Authors · 2025-11-27
>
> Dear Reviewer 7GqJ,
>
> I hope this message finds you well. As the rebuttal discussion period is drawing to a close (with less than one week remaining), we wanted to follow up to confirm whether all your concerns have been adequately addressed in our previous response.
>
> If you have any remaining comments, additional feedback, or further points that require clarification, please do not hesitate to let us know. Your insights are invaluable to refining our work, and we are committed to addressing any outstanding issues to enhance the quality of the paper.
>
> Thank you sincerely for your time, diligence, and constructive input during the review process. We greatly appreciate your efforts in helping us improve the manuscript.
>
> Best regards,
>
> KBVQ-MoE  Authors

---

### Comment · Area_Chair_USSV · 2025-11-24

Dear Reviewers,

**We kindly encourage you to review and respond to the authors’ rebuttals**. Your timely feedback is important for ensuring a fair and thorough review process. Thank you for your contributions to ICLR 2026.

AC

---

### Meta-Review · Area_Chair_4rML · 2026-01-01

**Summary:**

Across the five reviews, the reviewers consistently recognized that this paper addresses an important and timely challenge in MoE post-training quantization, and they appreciated the novelty and strong empirical improvements demonstrated by the proposed IDRE and BCOS framework, especially in ultra-low-bit settings where existing methods often struggle. The contribution was viewed as practically valuable, conceptually well motivated, and closely aligned with MoE-specific structural properties such as cross-expert redundancy and error amplification.

Reviewers nevertheless raised several concerns that guided the evaluation. They requested broader and more rigorous experimental coverage, including additional reasoning-oriented and post-training benchmarks such as MMLU, GSM8K, and HumanEval, as well as more extensive comparisons against stronger nonlinear or MoE-targeted baselines under iso-compression configurations. System-level analysis was also noted as initially insufficient, with calls for clearer measurements of practical calibration costs and inference behavior on real hardware, given the claimed efficiency benefits of vector quantization in MoE architectures.

Some reviewers pointed out clarity and presentation issues, including typographical inconsistencies, notation clarity, and ambiguity in describing certain components. They also requested stronger justification for the design choices such as the shared-subspace rank selection and more detailed ablations to clarify the independent roles of IDRE and BCOS.

**Reviewer Concerns:**

After reviewing both the reviewers’ concerns and the authors’ rebuttal, I find that the majority of the key issues raised during the review process have been addressed very convincingly. The rebuttal includes extensive additional experiments and highly detailed explanations, which were extremely helpful in evaluating the technical soundness and practical relevance of the work.

The authors significantly broadened their empirical results to include complex reasoning and post-training benchmarks such as MMLU, GSM8K, MathQA, HumanEval, and LongEval. These new results demonstrate that the proposed IDRE and BCOS mechanisms preserve model capability across diverse and demanding evaluation settings, reinforcing the strength of the method.

The addition of comparisons against advanced nonlinear quantization techniques (VPTQ, GPTVQ, AQLM, QUIP series) as well as MoE-focused methods (SubMoE, D2-MoE, EAC-MoE) clarifies the novelty and MoE specificity of the contribution. This makes the improvement significantly more compelling than what was visible in the original submission.

Practical deployability concerns were addressed with concrete runtime cost and calibration analysis, confirming negligible latency overhead. The improved clarity in notation, the explicit explanation of which experts are affected by IDRE, and theoretical plus empirical justification for keeping the shared subspace in full precision all contribute to substantially enhanced readability and rigor.

The rebuttal also contains carefully structured ablation studies that isolate the contributions of KLT, SVD truncation, and mean–variance correction, which was essential to understanding the internal effectiveness of the proposed components. These additions greatly improved the transparency of the method.

At this stage, the remaining points are minor: a bit more polish on exposition and potentially deeper kernel-level performance profiling in future iterations. These do not meaningfully diminish the contribution.

Overall, the paper has become significantly stronger through the review process. The depth and thoroughness of the rebuttal were particularly notable and provided strong assurance that the proposed method is both novel and robust for MoE quantization under ultra-low-bit constraints. I am satisfied that the major technical and empirical concerns have been fully addressed, and the revised work now presents a compelling and impactful contribution to the field.

**Reviewer Scores:**

Based on my assessment of the rebuttal quality and how thoroughly the authors addressed all major concerns raised, I believe that most reviewers would likely have improved or at least maintained their original scores had they been able to participate in further discussion. The rebuttal offered extensive additional experiments, clear theoretical justification, and improved presentation quality, which directly aligned with reviewers’ requests.

For the reviewers who initially raised concerns about evaluation breadth, missing baselines, and the need for stronger justification of design decisions, the newly added results and detailed explanations seem sufficient to resolve those issues. Therefore, I anticipate that these reviewers would have positively adjusted their scores.

Across the board, the paper has clearly improved in both technical depth and empirical completeness. As a result, I believe the collective sentiment among reviewers would have shifted toward a more uniformly favorable assessment following a full discussion.

---

### Decision · Program_Chairs · 2026-01-26

Accept (Poster)